# Formation of high-aspect-ratio nanocavity in LiF crystal using a femtosecond X-ray free-electron laser pulse

Sergey S. Makarov [1] ✉, Vasily V. Zhakhovsky [1], Sergey Yu. Grigoryev[1], Petr Chuprov[2], Tatiana A. Pikuz [3], Nail A. Inogamov [1,4], Victor A. Khokhlov [1,4], Yuri V. Petrov[4], Evgeniy A. Perov[1], Vadim Shepelev[2], Takehisa Shobu[5], Aki Tominaga [5], Ludovic Rapp [6], Saulius Juodkazis [7,8], Mikako Makita[9], Motoaki Nakatsutsumi [9], Thomas R. Preston [9], Karen Appel [9], Zuzana Konopkova[9], Valerio Cerantola[9,10], Erik Brambrink[9], Jan-Patrick Schwinkendorf[9], István Mohacsi[9], Vojtech Vozda [11], Vera Hajkova[11], Tomas Burian [11], Jaromir Chalupsky[11], Libor Juha [11], Norimasa Ozaki [12], Ryosuke Kodama[12,13], Ulf Zastrau [9], Andrei V. Rode [6] & Sergey A. Pikuz [14]

Recent research and development into the formation of nanoscale channels as a central component of nanofluidic biochip systems revolutionized the biological and chemical fields. Exploration of new pathways to form nanochannels is increasingly necessary to provide a new generation of analytical tools with accurate control of liquid fluid flow, high selectivity and increased mass flow rate. Here, we demonstrate that a single 9-keV pulse from X-ray free-electron-laser can form a nanoscale mm-long cavity in LiF. The laser-generated shock pressure results in channel formation with >1,000 length-to-diameter aspect ratio. The development of void is analyzed via continuum and atomistic simulations revealing a sequence of processes leading to the final long cavity structure. This work presents the study of mm-long nanochannel formation by a single high-brilliance X-ray free-electron laser pulse. With MHz repetition rate X-ray free electron laser opens a new avenue for the development of lab-on-chip applications in any material, including those non-transparent to optical lasers.

The fast-expanding area of research into the search for new ways to form nanochannels with high, more than 1000 length-to-diameter aspect ratios, is driven by the request to provide a new generation of nanofluidic analytical tools for biological applications[1–3]. Analytical nanosystems composed of arrays of closely packed nanochannels hold promises to provide accurate and highly selective molecular transport, ultra-sensitive analysis at a single-cell and single-molecular level[4–6]. The small, nanoscale-level sample volumes for diagnostics, applications such as DNA mapping and separation, the increased demand for high throughput screening technologies, and advanced lab-on-chip technologies are the major driving forces for the search for new cost-effective fabrication techniques for achieving smaller channel cross-sections, higher precision, and device-to-device reproducibility[2,5–7].

From a fundamental perspective, chemical reactions, synthetic chemistry, and protein dynamics in nanochannels are different from their larger channel analogs. It has been widely reported that some transport properties of confined liquids demonstrate significant deviations from the bulk values, including diffusion coefficient, shear viscosity, and thermal conductivity[8,9]. The unique features of unusual liquid flow in nanochannels include nonlinear transport due to a

surface charge at the interface of the solution and the nanochannel wall, Coulomb blockade, and ultrafast water transport[10,11]. New electrochemical phenomena are emerging for investigating fundamental physics and chemistry principles in the nanoscale, new fluid physics, and new principles that govern transport in nanochannels[10,12,13].

Among the various nanochannel fabrication technologies, ultrafast laser processing is a promising tool for the fabrication of nanofluidics due to its flexibility, versatility, high fabrication resolution, and three-dimensional fabrication capability[14–16]. Ultrafast lasers are a category of pulsed lasers with ultrashort pulse duration from a few picoseconds ($1\,ps = 10^{-12}\,s$) down to several tens of femtoseconds ($1\,fs = 10^{-15}\,s$). Due to very short pulse duration, they have high peak intensities that can modify transparent materials by multiphoton absorption and formation of voids inside the bulk of the material by strong shock and rarefaction waves, which are impossible to achieve with any other technologies. In addition, femtosecond laser pulses irradiating solid-state matter offer the unique capability to create nonequilibrium states, which allows structure manipulations of solid materials beyond thermodynamic limits[17–22].

It was first demonstrated that nanochannels with a high length-to-diameter aspect ratio could be generated via microexplosion conditions with femtosecond pulses converted into a needle-like non-diffracting Bessel-Gauss beam[23]. In essence, the Bessel-Gauss beam is a cylindrically symmetric interference field created by the coherent superposition of cone-shaped optical plane waves[24]. The Bessel-shaped pulse transforms large amounts of material by creating nanochannels with a very high aspect ratio ~1:100[17,23]. Due to the cylindrical geometry of the shock wave expansion, it creates a higher energy density than previously achieved with a spherically expanding shock wave generated by tightly focused Gaussian beams[17,23,25]. However, as in the case of microexplosion with Gaussian beams, formation of the Gauss-Bessel beams in the bulk of a solid is possible only in transparent materials for the laser wavelength to attune the required MJ·cm⁻³ (or >TPa pressure, $1\,MJ\cdot cm^{-3} = 1\,TPa$) level of energy density above the Young modulus, the characteristic of material strength[26,27]. Despite the progress made in nanochannel formation with ultrafast Bessel-shaped ultrashort laser pulses to date, the fabrication of nanochannels with controlled sizes and well-controlled surface properties responsive to fluidic transport is still a challenge. The major limitation is the ability to form Bessel-shaped beam profiles only in the bulk of transparent materials for optical pulses.

The advent of X-ray free-electron laser facilities (FEL) opens up unique opportunities for studying the effects of ultra-intense femtosecond XUV/X-ray radiation on various materials to determine the threshold of radiation resistance, model their decomposition, and secure further development technologies. A key advantage of these lasers for the direct fabrication of nano-structures is the unique combination of exceptionally high photon energy, which offers deep penetration into the bulk of any material, high spatial coherence, which guarantees very high directionality and focusing of the X-ray pulses to nanoscale beam cross-section, and high peak power to enable high energy density deposition deep into the bulk of potentially any material[28]. Thanks to the development of technologies for focusing XFEL beams to a spot of several tens to hundreds of nm with a pulse energy of several tens to hundreds of µJ, combined with a few tens of femtosecond pulse duration makes it possible to achieve pulse intensity in the range of ~$10^{18}$–$10^{22}$ W·cm⁻² [29–33], which is already comparable to modern petawatt laser systems operating in the optical spectral range[34].

Here, we present the results on the formation of a nanocavity with a submicron diameter and a high, more than 1000 times, aspect ratio in the bulk of the LiF crystal by focusing a single 20-fs, 9-keV X-ray pulse from the XFEL laser. Such a structure was observed at the energy density more than two times larger than previously reported values for a single X-ray pulse damage fluence threshold of ~190 J·cm⁻² for this

material[35]. LiF is often used as a fluorescent X-ray detector, which is ensured by the possibility of creating color centers (CCs) when exposed to electromagnetic radiation with a photon energy of more than 14 eV[36–38]. The low atomic mass of LiF allows deep penetration of X-ray photons into the sample ($d_{att} = 475\,\mu m$ for 9 keV) and precise imprint of the X-ray beam with submicron spatial resolution observed by fluorescence optical microscope.

The results of our experimental studies, supported by the simulations, show that nanochannels with a large length-to-diameter ratio can be formed by a single shot from a high-intensity X-ray FEL with sufficiently high photon energy. Similar results can also be formed in metals, semiconductors, ceramics, polymers, virtually—in any optically opaque or transparent materials which is beyond the reach of optical lasers.

## Results

### Experimental method

The experiment was conducted at the high-energy-density (HED) instrument of the European X-ray Free Electron Laser (EuXFEL)[39]. An X-ray beam with a photon energy of 9 keV ($\lambda = 0.138\,nm$) and a duration of ~20 fs (full width at half maximum (FWHM)) was focused through beryllium compound refractive lenses (CRLs) into a spot with a size of $d_{FWHM} = 0.41\,\mu m$, see Fig. 1a. A more detailed description of the focal spot size measurements for this experiment is given in ref. 29. The target was a circle LiF crystal with a diameter of 20 mm and a thickness of 2 mm, and was put at the point of best focusing of the beam. Two regimes of single-pulse irradiation of a LiF crystal were investigated: $E_1 = 26.7\,\mu J$ per pulse and $E_2 = 81\,\mu J$ per pulse (the energy of the pulse was modified with an attenuation filter), with the estimated errors of 10%. The resulting intensity of the X-ray pulses on the sample surface was correspondingly $1.0 \times 10^{18}$ W·cm⁻² and $3.1 \times 10^{18}$ W·cm⁻². Under these conditions, the absorbed energy density has the maximal values of $\xi_1 = 297\,kJ\cdot cm^{-3}$ and $\xi_2 = 895\,kJ\cdot cm^{-3}$, respectively (see "Methods" section for conversion details). The last energy density corresponds to 0.62 absorbed photons per cubic nanometer, which is about 100 times smaller than the number of F atoms in the LiF crystal at normal conditions. It is also worth noting that both of the energy densities are two orders of magnitude higher than the damage threshold of LiF crystal (~ 4 kJ·cm⁻³ per pulse), as determined in our previous work[35]. After each single-pulse irradiation, the crystal was moved in a direction perpendicular to the beam so that the next irradiation would fall on the fresh surface of the sample. Shots with strong attenuation (up to $10^{-6}$) were previously performed to obtain the intensity distribution inside the beam. Such an intensity obviously did not damage the crystal. We concluded that the XFEL beam had a Gaussian-like intensity profile in the investigated focal plane.

An analysis of the interaction area of the XFEL beam with the LiF sample was performed with three different readout systems (Fig. 1b, see "Methods" section for details). A scanning electron microscope (SEM) was used to study the damage on the surface, Fig. 1b (Step I). A confocal laser scanning microscope (LSM) in fluorescence mode was used to visualize the XFEL beam trace throughout the depth of the LiF, Fig. 1b (Step II). This is possible because the LiF crystal is a fluorescent medium, and when irradiated with X-rays, CCs are created in it, the distribution of which corresponds to the irradiation area[36,40,41]. For additional analysis of deep damage near the LiF surface, the well-known focused ion beam (FIB)−SEM technology[42,43] was used, etching layer by layer deep into the LiF sample (the first few tens of micrometers from the surface along the XFEL propagation axis) with simultaneous visualization of the fracture morphology by SEM (45/90° to the XFEL beam), Fig. 1b (Step III).

### Morphology of the generated cavity

First, the surface of the exposed LiF sample around interaction with the XFEL beam was analyzed with an SEM. Figure 2a–e shows the scan

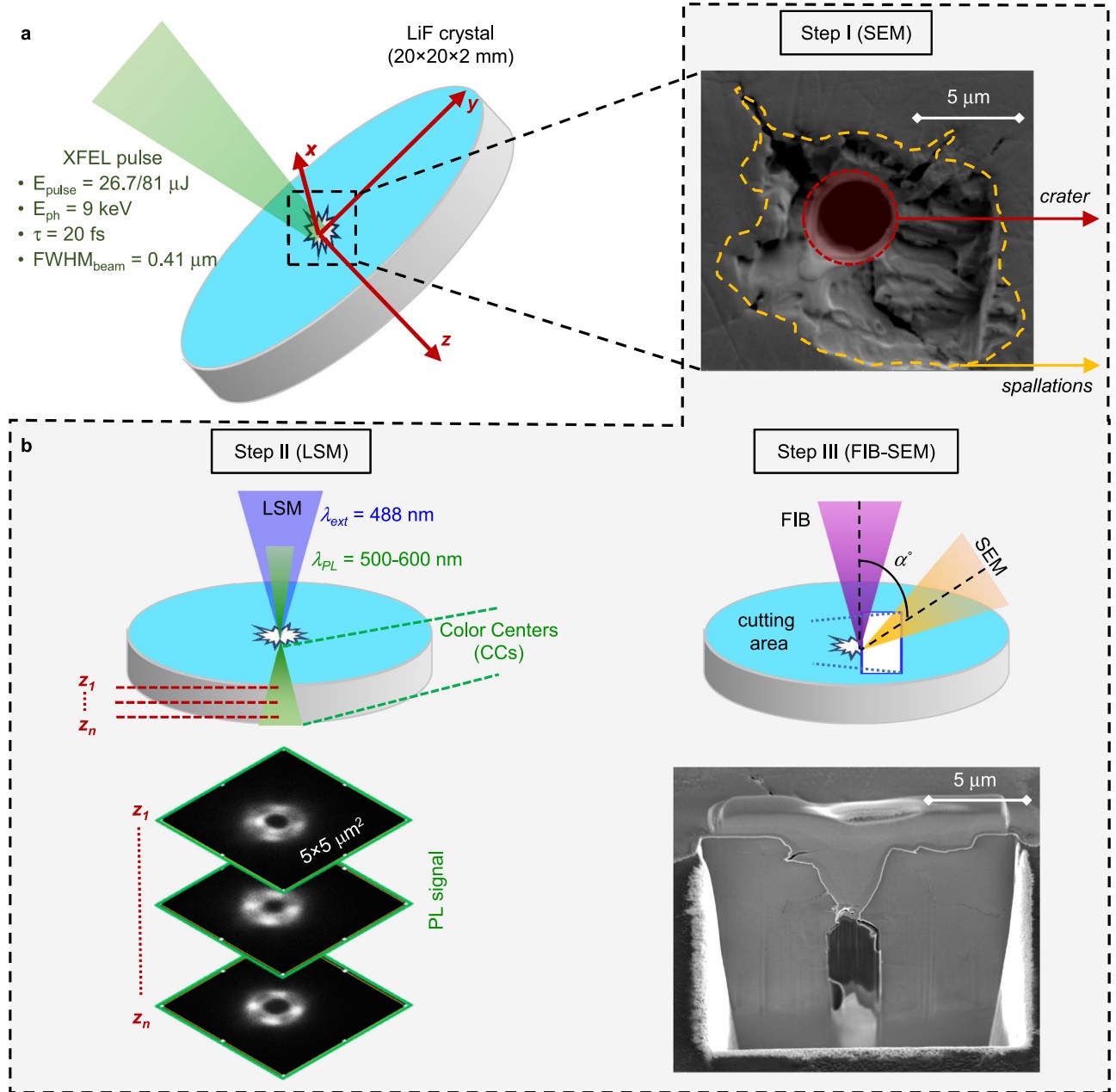

**Fig. 1 | Experiment to create a cavity in a LiF crystal with an X-ray FEL beam. a** Schematic view of LiF sample irradiation. **b** Diagnostics to study the morphology of the damage−step I (scanning electron microscope (SEM)), step II (reading the PhotoLuminescence (PL) signal with a laser confocal microscope (LSM)), step III (study of the internal morphology of the cavity by cutting the sample with a focused ion beam (FIB) and SEM illumination).

results after a single-pulse X-ray exposure with $\xi_1 = 297$ kJ·cm$^{-3}$ (a,b) and $\xi_2 = 895$ kJ·cm$^{-3}$ (d, e). As can be seen from the images obtained by scanning with an electron beam at an angle $\alpha = 45°$ to the LiF surface (Fig. 2a, d), in both cases there are two qualitatively similar structures: a crater with clearly visible bulges of frozen material along its edges (in purple), as well as, the area of delamination and spalling away from the influence of the X-ray beam. Viewing the damaged surface at an angle $\alpha = 90°$ (along the axis of incidence of the XFEL beam), Fig. 2b, e, reveals an additional feature in the damage pattern−a hole in the central area of the beam (burgundy area). The observed damage structure for two LiF irradiation regimes thus has the following features, Fig. 2b, e: Clearly visible hole in the area of maximum impact intensity $r_{\text{hole-1}} = 0.3 \times 1 \, \mu m^2$ ($r_{\text{hole-2}} = 0.4 \times 1.1 \, \mu m^2$)−burgundy area. The indices 1 and 2 refer to $\xi_1$ and $\xi_2$; cylindrical melting region around a crater with a radius $r_{\text{melt-1}} = 0.7 \, \mu m$ ($r_{\text{melt-2}} = 1.65 \, \mu m$)−purple region.

We refer to the annular area between the burgundy and dashed-purple contours as a crater. We emphasize that both the melting region and the region of the hole entrance have a round shape. They are therefore caused by plastic processes and melting, and not by a brittle fracture of the crystal. In this respect, these areas differ clearly from the area of the cracks; area of spalling of material with diverging cracks up to distances $r_{\text{cracks-1}} = 3 \, \mu m$ ($r_{\text{cracks-2}} = 10 \, \mu m$) from the center of the beam.

If we compare the characteristic sizes of the damage for two irradiation energy densities, it becomes clear that the size of the entry into the cavity is approximately the same. Along with the melting radius $r_{\text{melt-2}}/r_{\text{melt-1}}$ is ~2.2 times larger, the size of the crack area in $r_{\text{cracks-2}}/r_{\text{cracks-1}}$ is ~3.3 times larger, with the ratio of irradiation cases $\xi_2/\xi_1 = 3$.

The distribution of the absorbed energy density $\xi$ of the incident beam on the surface of the LiF crystal is shown in Fig. 2c, f for two

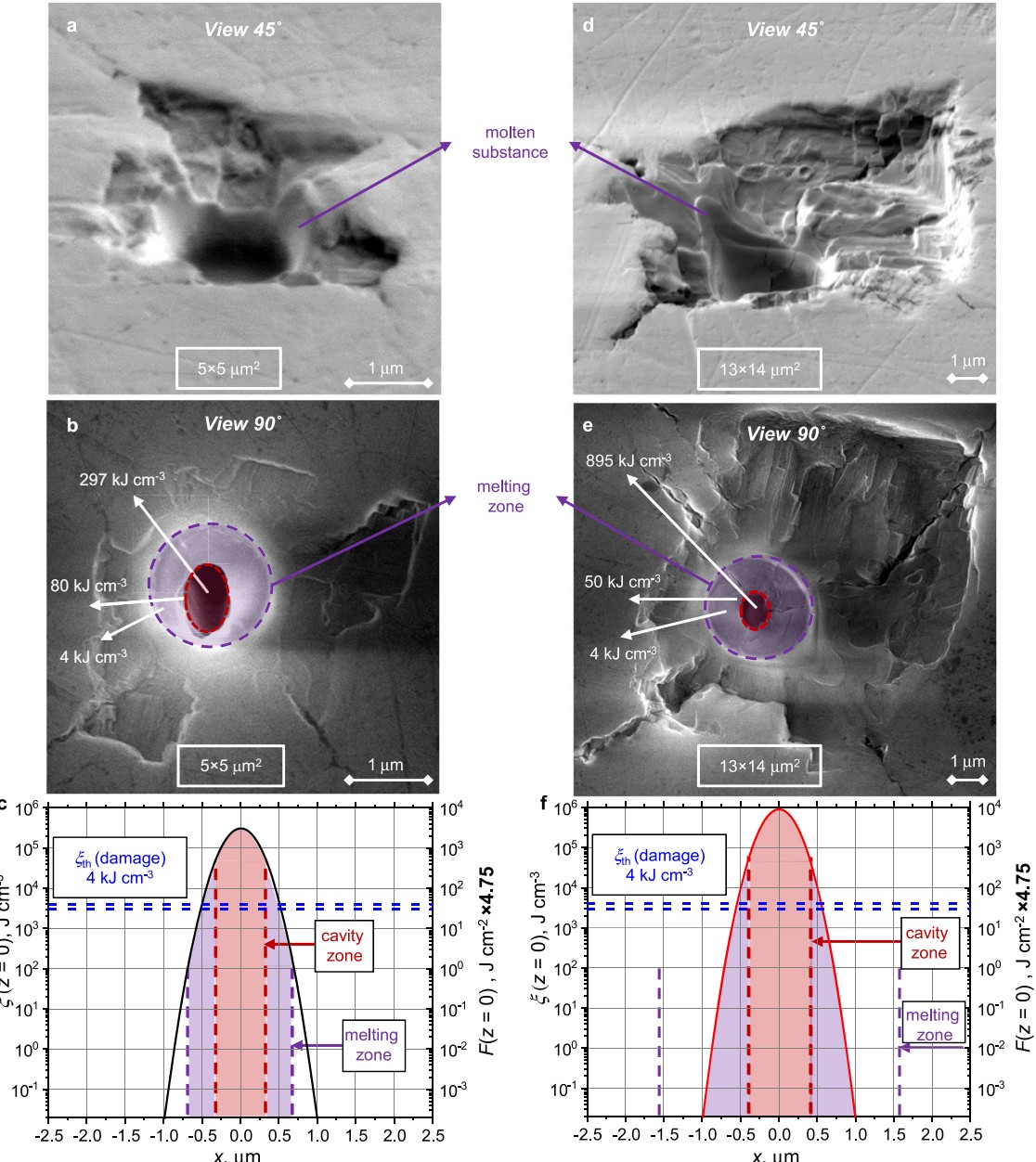

**Fig. 2 | Morphology of the surface damage of an exposed LiF sample.** SEM image when viewed at an angle of 45-90° to the crystal surface after irradiation with an absorbed energy density of **a**, **b** $\xi_1$ [26.9 μJ per pulse] = 297 kJ·cm⁻³ and **d**, **e** $\xi_2$ [81 μJ per pulse] = 895 kJ·cm⁻³. **c**, **f** Corresponding spatial distributions of the energy density on the surface of the LiF sample ($z = 0$). The purple and burgundy dashed lines correspond to the position of the corresponding regions highlighted in (**b**, **e**).

irradiation regimes (see "Methods" for calculation details). Note that the distributions are plotted on a logarithmic scale up to the value $\xi = 10^{-2}$ J·cm⁻³ that corresponds to the threshold for readout of the PL signal of $F_2$ and $F_3^+$ CCs with LSM[36,44]. As can be seen in Fig. 2c, f, the decrease of $\xi(x)$ is quite sharp (strong)—the level of intensity decreases $e$ times relative to the maximum of the Gaussian function at a radius $r_{1/e} = FWHM/2/\sqrt{2} = 0.296$ μm. It is worth paying attention to the value of the absorbed energy density $\xi$ at the boundaries of the characteristic structures (2–3). To cause spalling of the material (case 3), the value of $\xi$ must be higher than the elastic limit for LiF of ~ 4 kJ·cm⁻³ [35] [dashed-blue horizontal lines in Fig. 2c, f]. However, as shown in Fig. 2b–e, $\xi$ is much smaller than 0.01 kJ cm⁻³ in the region of cracks. Consequently, the pressures directly associated with a heating process (isochoric as it will be considered below) in these radii are much lower than 100 atm, and these amplitudes are completely insufficient for the brittle fracture of a LiF crystal. A similar remark applies to the melting

region (case 2) in Fig. 2b–e. It should be underlined that Young's modulus of LiF of 64.8 kJ·cm⁻³, which indicates pressure at 64.8 GPa, where the shock wave converts into an acoustic wave. Melting requires an increase in internal energy of about 6 kJ·cm⁻³ compared to the resting state at room temperature. However, at the edge of the melting area (dashed-purple contour), the local energy value is three orders of magnitude lower (-0.03 kJ·cm⁻³). Considering the above, the formation of regions (2–3) (melting region–crack region) is due to the attenuation of the shock waves (SW) generated by the influence of the XFEL beam on characteristic amplitudes. The initial amplitude of the shock wave is determined by the energy of the laser pulse, but the attenuation of the shock wave during propagation in LiF is much slower compared to the decrease in absorbed energy density.

At the next stage, the structure of the internal damage (deep in the sample) was investigated, namely the hole observed at the bottom of the crater (burgundy area in Fig. 2b, e). Note that the use of AFM for

this purpose is not sufficient because, under our experimental conditions, the area affected by the beam with $d_{FWHM} = 0.41\ \mu m$ is too small and too deep to be fully inspected by a standard AFM tip. Therefore, to investigate the internal structure of the damage, we used a confocal laser scanning microscope with a layer-by-layer reading of the PL signal deep in the LiF sample (up to $z = 1200\ \mu m$) with a step $\Delta z = 1\ \mu m$ near the crystal surface and with a step $\Delta z = 10\ \mu m$ for $z > 20\ \mu m$ (see more details in Methods). The measurements were performed using the microscope Zeiss LSM700 in fluorescence mode. Figure 3a shows the sequence of PL images (on the right) for the case of $\xi_1 = 297\ kJ\cdot cm^{-3}$ and a sketch of observed internal structures at corresponding depths (on the left). Images show the presence of a crater with a central hole in depth up to $z \sim 2\ \mu m$. Intriguingly, after the hole closure (PL signal is clearly seen at $z \sim 2$–$8\ \mu m$, indicating the presence of matter in this region), in deeper layers the PL signal disappeared again, and it was absent down to $z \sim 1000\ \mu m$. Thus, a long, narrow channel is formed along the beam propagation through the LiF sample. In the sketch that interprets the obtained images, three main zones are respectively highlighted: (zone 1) is a crater and a waist in the form of a plug, (zone 2) is a cavity in the form of a narrow, deep channel, and (zone 3) is the closure of the cavity. Note that since the primary analysis of the sample surface was done with the SEM, we see a bright PL signal outside the

X-ray beam region in the first layers of the images (Fig. 3a–$z = 2$–$8\ \mu m$), which is explained by the excitation of CCs created by the electron beam on the surface.

In Fig. 3b, the measurements of the transversal size of the channel vs the depth are presented for both irradiation cases $\xi_1$ (in black) and $\xi_2$ (in red). We can see that after a single pulse irradiation by the beam with $d_{FWHM} = 0.41\ \mu m$ и $\xi_1 = 297\ kJ\cdot cm^{-3}$, the damage of LiF has a complex structure with the following characteristic dimensions: crater ($\sim 2\ \mu m$), layer of matter ("plug" $\sim 7\ \mu m$), an ultra-deep channel (length $L_c \sim 1000$–$1100\ \mu m$) with a diameter $d_c \sim 0.6\ \mu m$ throughout the entire depth.

The channel formed in LiF has a very high aspect ratio ($d_c/L_c \sim 1/1600$–$1/1800$). A similar restructuring of material structure was observed for the irradiation regime with $\xi_2 = 895\ kJ\cdot cm^{-3}$, Fig. 3(b-red dots), but with a slightly larger channel width and length. Because the working distance was limited to 1.1 mm with the ×100 objective used, we were not able to observe the depth of the channel's end, so the aspect ratio is in fact even higher in these experiments.

Figure 3c shows the dependence of the decay of the energy density $\xi$ within the LiF crystal for two irradiation modes ($\xi_1 = 297\ kJ\cdot cm^{-3}$ — black curve, $\xi_2 = 895\ kJ\cdot cm^{-3}$—red curve) (calculation details in "Methods" section). One can see that the value of

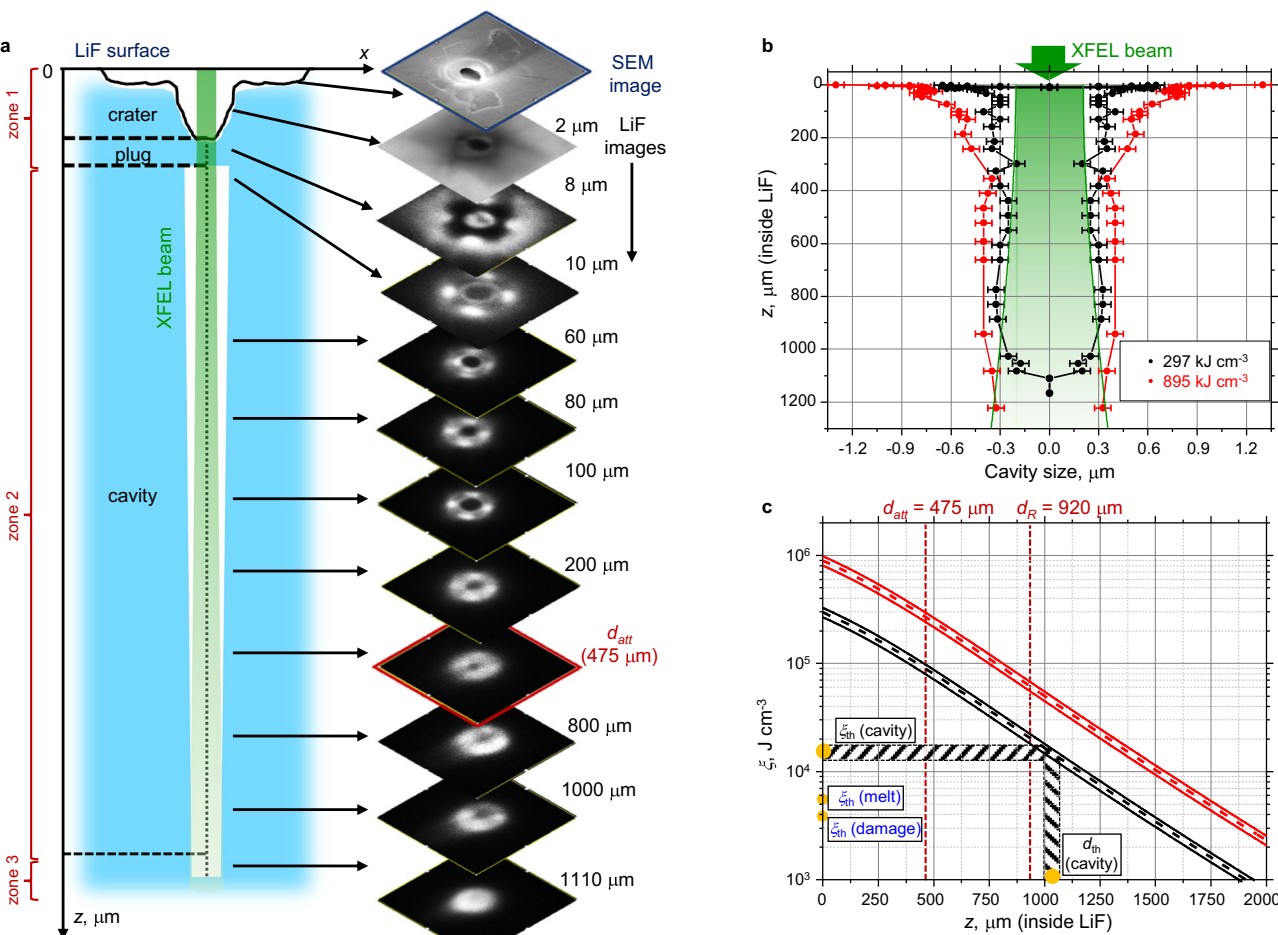

**Fig. 3 | Investigation of the internal structure of LiF destruction using a confocal microscope. a** Readout data of an exposed LiF crystal at an absorbed energy density $\xi_1 = 297\ kJ\cdot cm^{-3}$. The readout was performed with an LSM700 confocal fluorescence microscope deep inside the crystal in the range $z = 0\ \mu m$ (surface) to $z = 1200\ \mu m$. The identified areas within the sample are shown schematically on the left. **b** Cavity dimensions measured from PL images as a function of LiF scan depth for exposures with $\xi_1 = 297\ kJ\cdot cm^{-3}$ (black dots) and $\xi_2 = 895\ kJ\cdot cm^{-3}$ (red dots). The average error in measuring the cavity diameter was 50 nm, and determined by the

instrumental error of the microscope. The green shading shows the region of propagation of the XFEL beam deep in LiF, considering the divergence (calculation details in Methods). **c** Distribution of the absorbed energy density $\xi$ within the LiF crystal at different depths $z$ (in the direction of the XFEL beam) after a single pulse: the black line corresponds to $\xi_1\ [z = 0] = 297\ kJ\cdot cm^{-3}$ at the surface, and the red line corresponds to $\xi_2\ [z = 0] = 895\ kJ\cdot cm^{-3}$. The deviations from the mean value were determined by a 10% value corresponding to the estimated error of X-ray pulse energy.

$\xi_l[z = 1000{-}1100\ \mu m] = 12{-}18\ kJ \cdot cm^{-3}$, where the channel ends, is ~16–25 times lower than at the surface $\xi_l[z = 0\ \mu m] = 297\ kJ \cdot cm^{-3}$. The experimentally determined channel formation threshold is $\xi_{th} \sim 3{-}4$ times higher than the threshold for the onset of LiF dielectric damage found in our previous work[35].

To verify that the observed absence of the PL signal in the LiF images in the central region of the XFEL beam (Fig. 3a) is due to the absence of matter (formation of a cavity), we applied FIB-SEM technology to open the expected channel layer by layer. Beginning at the crater area, the ion beam ablated the LiF sample layer by layer to a depth of several tens of micrometers from the surface. Simultaneously, the SEM visualized the fracture morphology at an angle of 45° to the sample surface (see Fig. 1b, Step III and "Methods" section for details). The LiF surface was preliminarily coated with the protective layer of platinum. Figure 4 shows the results of the corresponding SEM measurements after crystal etching for an exposure case $\xi_2 = 895\ kJ \cdot cm^{-3}$. Both in a wide field of view (Fig. 4a) and with a greater magnification image (Fig. 4b), the existence of a cylindrical cavity under the crater bottom is clearly seen as a black region. Also, Fig. 4b emphasizes details near the joint of the crater with the crater-initial channel area. The crater has the shape of a cone with a base size of ~3.3 μm, which is consistent with the melting area $r_{melt\text{-}2} = 3.3\ \mu m$ in Fig. 2e, f. The images in Fig. 4a, b) show a similar fracture structure as the PL images in Fig. 3a: (1) crater (cone-shaped), (2) thin substance layer (plug), and (3) expanded cavity. According to the SEM images for the case of exposure $\xi_2 = 895\ kJ \cdot cm^{-3}$, Fig. 4b, the cavity width decreases from 2.8 to 2 μm at depths up to $z = 21\ \mu m$, which also corresponds to the measurements of this value from the PL images in Fig. 3(b–red dots). At greater depths, the channel narrows to a radius of 0.4 μm and its size remains practically unchanged, see Fig. 3(b–red dots).

The experimental results on the formation of a deep cavity with an ultra-high aspect ratio inside the LiF material require a theoretical explanation, as such a phenomenon has not been previously observed.

## Multi-method simulations

The aim of the numerical simulation is to relate the energy release of the XFEL beam to the features formed (crater, plug, cylindrical cavity radius, and channel floor). Since the formed structure has a complex morphology and a high aspect ratio (see Fig. 3a—zones 1–3), a quantitative description with a single model and one computational code cannot describe the whole physics of the formation of such an elongated structure. Taking into account our simulation results obtained for various energy densities deposited at the different depths, the LiF sample was divided into the following (see Fig. 3a): (zone 1) where the crater and the substance plug are produced by superposition of the cylindrical rarefaction wave with the unloading waves propagated from the free surface (3D SPH and 2D Hydrodynamic (HD) simulations); (zone 2) includes the most part of long cavity produced by the cylindrical rarefaction wave (2D SPH and molecular dynamic (MD) simulations); (zone 3) where energy deposition is near a threshold of cavity formation. Here, the cavity is produced by cavitation in the stretched melt, and the cavity floor is formed by crystallization of the melt (MD).

We should underline that the phase transition into a plasma state happens within a femtosecond at the very early stage of the pulse. The 20-fs pulse intensity is at the level of $3.1 \times 10^{18}\ W \cdot cm^{-2}$, which is far below the criterion for the nonlinear saturable absorption of 9-keV photons to emerge in solids (as an example, it was demonstrated in refs. 45,46 for Fe). Due to the long attenuation length, the maximal absorbed energy density near the sample surface is $\xi_2 = 895\ kJ \cdot cm^{-3}$ for such a laser pulse, which corresponds to the concentration of absorbed photons $0.62\ nm^{-3}$ is about 100 times smaller than the concentration of F atoms in the LiF crystal (61 atoms $nm^{-3}$). For this reason, we assume a linear absorption process for our experimental conditions. The formation of the cavity does not depend on a detailed description of the almost isochoric formation of the plasma in a very short time (~fs), but only on the profile of the input energy. The stage of formation of the hot zone with plasma (i.e., the processes of ionization, deceleration of fast electrons and radiative transfer) is not important for the description of the formation of the cavity, since the radius of the hot zone is known from the experiment and we use the density of deposited energy (not incident) to determine the initial pressure and temperature from the equation of state for modeling the motion of matter leading to the formation of the cavity. In the following simulations, we used the distribution of the energy density inside the LiF crystal, as shown in Fig. 3c.

To simulate the dynamics of cavity opening, as well as damage to lithium fluoride as a result of irradiation, the LiF model implemented within the framework of the smoothed particle method[47,48] (see details in Methods) is used as in our previous work[35]. Figure 4c shows the simulated curves of the cavity radius $R$ and the radial velocity of its boundary $v_r$ as a function of time. The diagram shows that the cavity expands and reaches its maximum size in the first few ns after the start of irradiation. At time $t \sim 3\ ns$, the growth of the cavity practically stops, as can be seen from the velocity curve $v_r(t)$. After that, its size remains almost unchanged, as it is observed in the experiment.

The contraction of the cavity begins after the arrival of a rarefaction wave from the free boundary of the target, which reduces the pressure inside the cavity to zero. To account for the effect of this rarefaction wave in our 2D simulations, the pressure inside the cavity starts to drop after a certain time, corresponding to the depth of the simulated layer. In our simulations, the pressure drop occurs within ~10 ns. After the pressure has dropped to zero, elastic forces around the cavity lead to a reduction in the channel radius. The diagram in Fig. 4c shows that in the time interval $t = 10{-}20\ ns$, the radial velocity $v_r(t)$ becomes negative, and at the same time, the radius of the cavity $r(t)$ begins to decrease sharply. After the pressure inside the cavity has completely dissipated, at $t > 20\ ns$, the cavity continues to narrow due to inertia, but at a much slower speed. The diagram of the radial velocity $v_r(t)$ shows that it approaches zero at $t > 30\ ns$, i.e., the change in cavity size practically comes to a standstill. From this moment onwards, the radius of the cavity tends towards its asymptotic value.

Figure 4d shows a comparison of the cavity radius as a function of depth, which was determined in the experiment and in the simulation in irradiation mode $\xi_2 = 895\ kJ \cdot cm^{-3}$. In the graph, the gray markers correspond to the simulated maximum cavity radius reached in the expansion stage of the cavity, the black markers correspond to the final cavity radius in the simulations, and the red markers correspond to the experimental results. Firstly, there is a good quantitative agreement between the simulations and the experimental data. Secondly, one can see from the above diagram that in the simulations, the final size of the cavity is approximately the same at different depths (it decreases slightly with depth), although the maximum achievable size varies greatly. The experiment also shows a similar picture: the transverse size of the cavity remains practically unchanged over the entire depth of the channel. A noticeable widening is only observed near the free surface.

The insets (e2, e3) to Fig. 4 show the images of LiF damage obtained in the simulations for the irradiation mode $\xi_2 = 895\ kJ \cdot cm^{-3}$ at different distances from the irradiated surface of the sample—depth 390 μm (e2) and 660 μm (e3), respectively. The gray color in the figures represents the intact material, and the black color corresponds to the damaged area. The central part, shown in white, is a cavity. From the pictures above, one can see that the damaged area decreases with depth. This is quite natural, as the specific energy absorbed and, therefore, the amplitude of the shock wave decrease with depth. Here, the material is damaged along clearly defined damage bands, the concentration of which gradually decreases with the distance from the area of initial heating. The appearance of cracks is clearly observed around the nanochannel. The cracks propagate up to 5 μm away from

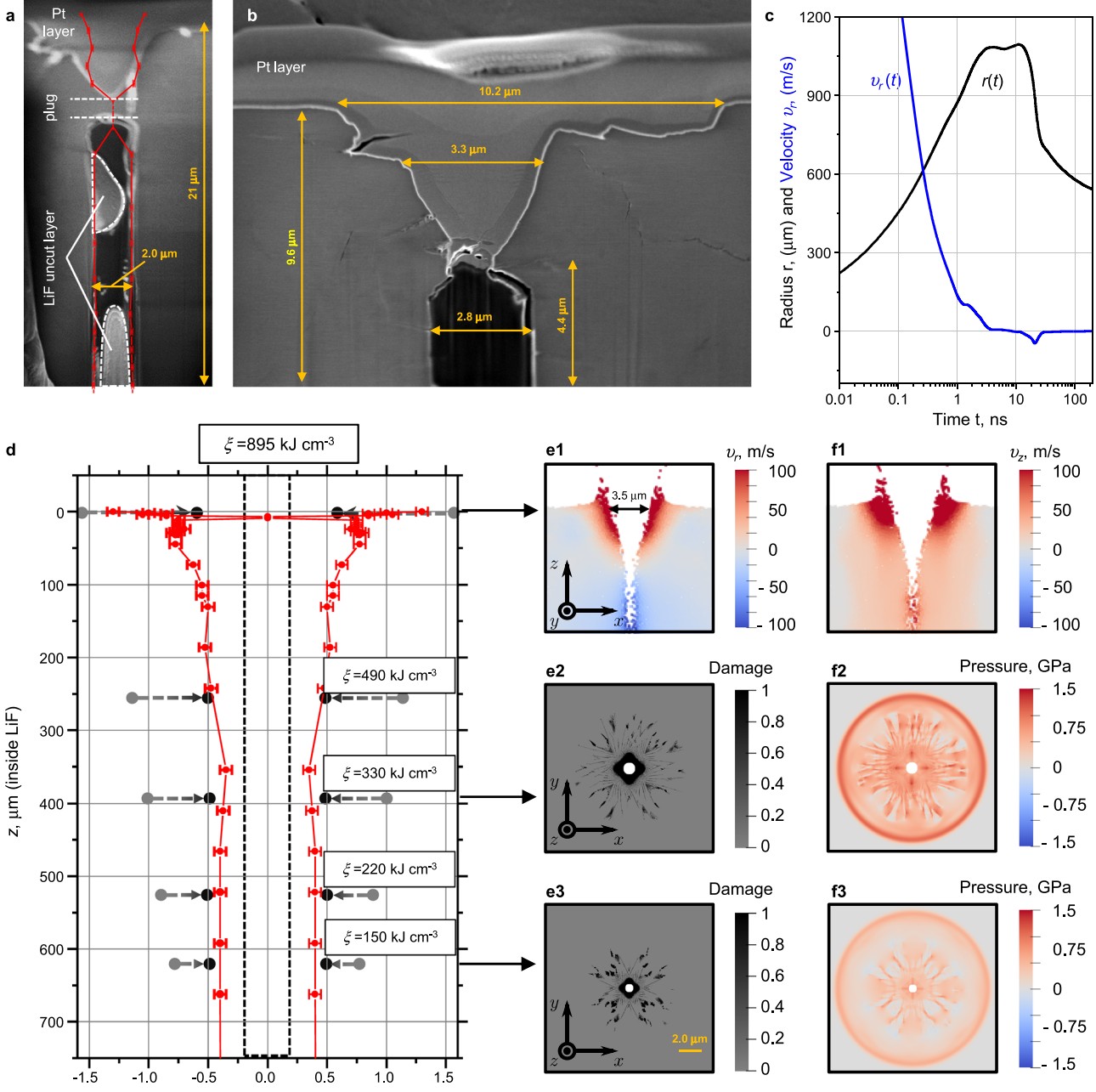

**Fig. 4 | FIB-SEM measurements and comparison with smooth particle hydro-dynamic (SPH) simulation.** Irradiation case $\xi_2 = 895\ \text{kJ·cm}^{-3}$. **a** Wide scan area near the crystal surface (in red, the PL measurements from Fig. 3b are shown). **b** Narrow area at the "crater-graft-beginning of the cavity" boundary (the average error in measuring the cavity diameter was 50 nm and determined by the instrumental error of the microscope). **c** Modeling results for evolution of the cavity radius $r(t)$ and the radial velocity of its boundary $v_r(t)$ at a depth of 390 µm (2D code). **d** Dependence of the final cavity size on the depth: red dots correspond to experimental data (the average error in measuring the cavity diameter was 50 nm and determined by the instrumental error of the microscope); black and gray dots correspond to simulation results (2D code). The initial cavity size given in the simulations is indicated by two vertical dashed-black lines. The gray circles correspond to the maximum radius to which the cavity expands in the simulations. **e1, f1** Simulated fields of radial $v_r$- and axial $v_z$-velocities formed in the $xz$-plane near the surface (3D). **e2, e3** The LiF damage patterns formed at a depth of 390 and 660 µm at time $t \simeq 1.75$ ns after the start of irradiation (2D code). The corresponding pressure field is shown on the right in the insets (**f2, f3**). The size of the images is given on a scale of $15 \times 15\ \mu\text{m}^2$.

the channel, and their appearance gradually decreases with the depth of the channel.

To analyze the dynamics of the cavity opening near the surface sample, an additional SPH simulation was performed in a 3-dimensional environment. In contrast to the expansion of the cavity into the depth, here, the type of deformation is mainly determined by the superposition of the cylindrical rarefaction wave with the unloading stress wave propagated from the free surface of the sample. In the insets (e1, f1) to Fig. 4, the radial $v_r$ and axial $v_z$ velocities determined in the 3D SPH simulation are shown in a longitudinal section. From the above figures, one can recognize the beginning of the formation of a cone-shaped crater similar to the one observed in our experiments (see Fig. 4b). From the velocity distribution $v_r$ shown in Fig. 4e1, it is clearly visible that the crater continues to expand near the surface, while at depth the cavity has already started to contract (the blue color in the lower part of the figure (e1) corresponds to the negative radial

velocity). The underestimation of the final radius determined in 2-dimensional simulations for the case of cavity expansion near the surface (Fig. 4c), therefore, seems reasonable, as the influence of the free surface was not considered. In a 3-dimensional simulation, where such an influence should be considered naturally, the radius of the crater is many times larger than the radius of the cavity at depth.

It is also noteworthy that the damage and pressure distributions determined in the simulations have a cross-shaped pattern similar to the experimental images of a confocal microscope, see Fig. 3a and Fig. 4e1, e2, f2, f3. The reason why the damaged area forms a structured cruciform shape in the simulations is that the SPH particles initially occupy the nodes of a close-packed 2-dimensional lattice. Thus, there are special directions along which the damage prefers to propagate in the LiF crystal. It was confirmed by a test run with a homogeneous liquid-like packing of SPH particles, which provides angle-independent damage and pressure distributions. A similar picture emerges in the experiment since the LiF sample contains preferential directions related to the lattice orientation of the investigated crystals.

Finally, to determine the threshold for crater formation and cavity opening, two-dimensional high-spatial resolution simulations were carried out using a 2D HD code for absorbed energy densities $\xi$ from 10.5 kJ·cm$^{-3}$ to 28 kJ·cm$^{-3}$ with a step of 3.5 kJ·cm$^{-3}$ (see Supplementary Information I). It was found that the first signs of the formation of a surface structure leading to the formation of a crater were observed at value $\xi_{crat} = 14$ kJ·cm$^{-3}$, which is generally consistent with the results of our early experimental work[35]. The calculated threshold value for cavity opening was $\xi_{cavity} = 28$ kJ·cm$^{-3}$, which is slightly higher than that observed in Fig. 3c (12–18 kJ·cm$^{-3}$).

To explain the mechanism of formation of the cross-shaped structure visible in the PL images (see Fig. 3a) around the cavity at the sample depth $z > 8$ μm, the large-scale MD simulation of the radial material movement in a deep slice of LiF crystal perpendicular to a rapidly heated cylindrical channel was performed (see "Methods" for more details). Since even in a large-scale MD setup, the sample size is limited by a micrometer, our MD simulations are intended to obtain a qualitative description of basic processes leading to the formation of a cavity in the depth of the LiF crystal heated to peak temperatures and

pressures much lower than those studied above, with the usage of the SPH method.

Simulation results presented in Fig. 5a were obtained long after fast heating of a cylindrical sample with a radius of 300 nm (in $XY$ plane) and thickness of 8.1 nm along the $Z$-axis with periodical conditions. The heating was performed by the Langevin thermostat with a characteristic time of 20 fs using a target Gaussian-like temperature profile in a central spot with a radius of 30 nm. Material melted up to 23 nm. The peak temperature of 19,000 K and pressure of 67 GPa are generated at the central axis, as seen in Fig. 5a1, where profiles of density and pressure are shown for a few early times after heating. See the full-scale simulation in the corresponding Supplemental Video 1.

The high pressure in the central hot spot (channel cross-section) generates a diverging cylindrical shock and a converging rarefaction wave, which both cause a radial flow of material, leading to a decrease in plasma pressure and temperature within the channel. Soon, the diverging shock wave splits into a fast elastic shock (precursor) and a slow plastic shock wave, see Fig. 5a1. As in the SPH simulation, the surrounding cold crystal is damaged by this plastic shock wave until it weakens enough. Due to the face-centered cubic (fcc) crystal structure of LiF, four corner damage sectors are formed around the radial cracks, which can be seen in Fig. 5a, similar to those observed in NaCl[49]. After traveling a sufficient distance from the central axis, the diverging shock wave attenuates and becomes purely elastic. Then the irreversible damage ceases, and the cold solid begins to elastically resist the radial outflow of material from the channel. Soon, the flow velocity reduces to zero, and the remaining pressure of about 1–2 GPa in the hot fluid within the channel comes to the mechanical equilibrium with the tightening elastic stress in the surrounding cold solid. Thus, at the given relatively low energy deposition, the cylindrical cavity is still not formed in the channel filled by hot fluid with 2–3 times less density than in the solid state, see Fig. 5a1. At this point, the rapid (acoustic) stage of material transformation ends, and a long cooling stage of the channel fluid, leading to liquid-vapor phase separation and crystallization of the melt, begins (see the corresponding discussion in Fig. 5b below). It is worth noting that our MD simulations performed at higher energy deposition show cavity formation at the end of the rapid stage if the peak pressure exceeds ~100 GP, at which the rarefaction wave

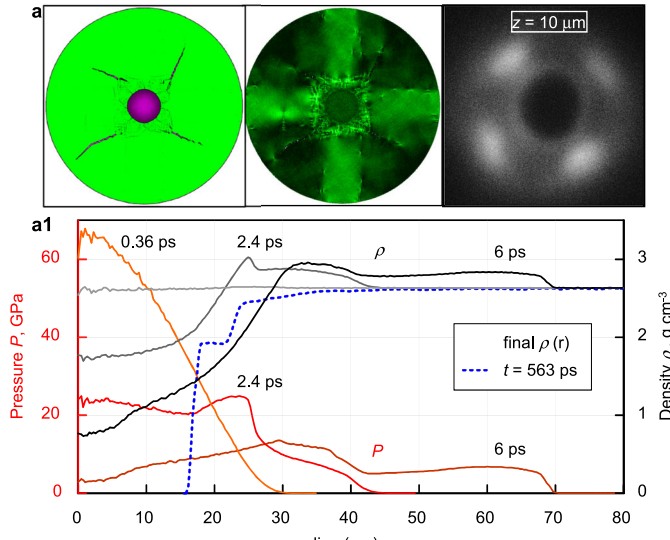

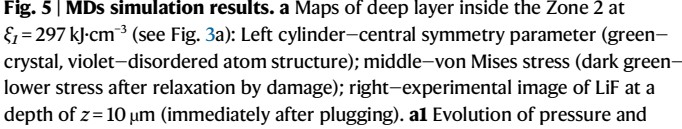

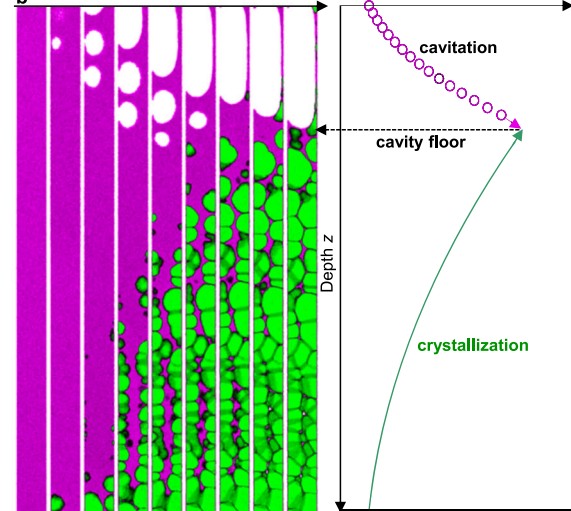

**Fig. 5 | MDs simulation results. a** Maps of deep layer inside the Zone 2 at $\xi_1 = 297$ kJ·cm$^{-3}$ (see Fig. 3a): Left cylinder−central symmetry parameter (green− crystal, violet−disordered atom structure); middle−von Mises stress (dark green− lower stress after relaxation by damage); right−experimental image of LiF at a depth of $z = 10$ μm (immediately after plugging). **a1** Evolution of pressure and

density profiles during several ps of acoustic stage. The final density profile with a cavity (blue line) is formed much later. **b** Simulations for $\xi_1[z = 1100$ μm$] = 12$ kJ·cm$^{-3}$, corresponding to the level $z$ ⁓ 1000 μm where the nanochannel ends (see zone 3 in Fig. 3a). Full-scale simulation is presented in the corresponding Supplemental Video 2.

generates high enough radial velocity to produce a cavity filled by a hot gas for a short time.

Figure 5a shows the 2D distributions of the damage and the equivalent von Mises stress after the damage has ceased, as well as the experimental PL image at a depth of $z = 10\,\mu m$. It is important to underline that the bright, cross-shaped PL signal cannot be caused by the XFEL beam being angularly uniform about its axis. Such an angular dependence of the concentration of CCs in LiF could arise under the influence of a sufficiently strong shock wave that creates four sectors with cracks, where the equivalent (shear) stress is reduced. Outside these sectors, the material behaves elastically, as shown by the cruciform area of high residual shear stress in the middle of Fig. 5a, where the CCs previously created by the XFEL beam can be erased. Thus, the concentration of CCs should remain high in four spots according to the experimental distribution in the right Fig. 5a. The observed increased brightness of the thin rim around the cavity is probably due either to rapid crystallization of the melt on the cold channel walls or to a very high concentration of damage leading to amorphization of the crystal.

We have also performed additional MD simulations of slow processes leading to the formation of the cavity at relatively low energy deposition near the cavity floor (Fig. 3). Such processes are driven by the slow loss of heat from the hot melt filling the cylindrical channel into the surrounding cold crystal. At this late stage, the pressure in the channel decreases due to the cooling of the melt, but the radius of the channel remains almost unchanged as the elastic stresses developed in the surrounding solid material prevent the channel from narrowing. This means that the pressure in the melt can become negative even before it freezes, which creates the conditions for cavitation, leading to the formation of cavities inside the channel.

Simulation results presented in Fig. 5b were obtained during long-term cooling of molten LiF in a cylinder with a length of 1000 nm surrounded by rigid walls at a radius of 10.3 nm. The initial sample was obtained by melting solid LiF with a linear temperature gradient from 1300 K at the bottom of the cylinder to 1500 K at the top. The pressure of 1.8 GPa was established along the cylinder, which is close to that remained after the stop of material movement in the MD simulation of the early stage of evolution shown in Fig. 5a. Then the Langevin thermostat was applied to $x$- and $y$-velocity of atoms to cool the initial temperature distribution by 900 K to a target slope of 300–500 K along $Z$-axis with the characteristic cooling time of 200 ps. Figure 5b shows only part of the sample near the cavity floor—see the full-scale simulation in the corresponding Supplemental Video 2.

Figure 5b (left) shows the processes of crystallization and cavitation in the cylindrical channel with the fixed solid walls, through which the heat flows simultaneously over the entire length of the channel. In the deep part, the melt has a lower temperature than in the upper part, so crystallization starts at the bottom and spreads upwards in the channel. The speed of movement of the crystallization front is controlled by the degree of undercooling of the melt and can, in principle, exceed the speed of sound if the heat is lost to the channel walls fast enough. However, the cooling rate is slow, so the crystallization process proceeds below the speed of sound. Cavitation begins when the negative pressure in the melt falls below the tensile strength of the liquid. This strength decreases with increasing temperature so that cavitation propagates from top to bottom along the channel towards the crystallization front—as shown in the schematic $z$–$t$ diagram in Fig. 5b(right). Our interpretation is that the sharp cavity boundary deep in the LiF crystal seen in Fig. 3b is caused by the cessation of cavitation upon meeting with the crystallization front. It is of interest to observe in Supplemental Video 2 that below the cavity floor, the fast homogeneous crystallization results in the formation of nanovoids between solid grains by reason of the much lower density of molten LiF compared to solid LiF (by 25%).

It should be noted that the simulated cavitation starts with independent nucleation of a few small voids in a tiny area, but soon they coalesce into a single bubble with a diameter of the channel of 20.6 nm. In experimental conditions, the size of the channel is about 600 nm, and cavitation may result in the formation of a foam-like material, which will decay, producing liquid filaments and droplets. Finally, such droplets will reach the cold cavity walls and be frozen, like observed in ref. 17. But this scenario is realized only if the walls are already in a solid state, i.e., at a large enough cooling rate of molten material in the channel. Otherwise, the walls should be smooth.

## Discussion

The effect of the formation of a deep cylindrical submicron cavity with >1000 length-to-diameter aspect ratio by an X-ray pulse from FEL was observed in our work. We emphasize that this effect is different from the classical ablation of material. The cavity is formed by the motion and deformation of a solid body in the bulk of the condensed target — i.e., the mass of the vapor/plasma ejected from the target into the vacuum is much smaller than the mass of the substance that filled the cavity before laser irradiation.

In general, there are three fundamentally different zones along the beam axis. This is the short first zone with a length of the order of a dozen beam diameters. The long second zone, with a length of the order of units of absorption lengths $d_{att}$, and the transitional zone, where the empty cavity ends. Its length is on the order of a fraction of the absorption length $d_{att}$. We underline that the cavity formation is caused by radial extrusion of the substance in the second zone. In this case, the substance is pushed radially in the direction away from the beam axis and crystallizes after its radial shift. The empty cavity arises mainly due to integral irreversible swelling of the target itself. The displacements of the macro-target boundaries are small because the size and volume (-1 cm$^3$, see Fig. 1) of the target are much larger than the cavity volume (of the order of $10^{-9}$ cm$^3$). It should be emphasized that the cavitation phenomenon takes place deep in the bulk of the target, i.e., at the end of the cavity in the third zone.

The results of this work can, therefore, be used to develop a method to create highly elongated, extended in depth multichannels with submicron diameters in a LiF crystal using femtosecond X-ray pulses. The method consists of "radial extrusion," a cavity in the sample with an XFEL beam, using the laser method by placing a focal point on the surface of the crystal. The surface morphology of the resulting hole can be observed with an SEM and by monitoring the internal structure of the cavity with a non-destructive method—using a confocal laser scanning microscope by recording the intensity of a fluorescent signal within the beam image.

It is worth noting that optical Gaussian and Bessel femtosecond beams are already being used to create nanochannels (even arrays of those) in solids[20,25,50,51]. The main limitation is the possibility of creating such structures only in optically transparent materials. An additional complication is introduced by the nonlinear light-matter interaction with a strong dependence of energy deposition on fluctuating local carrier density. This aspect limits the possibility of controlled creation of nanochannels. In our experiment, the aspect ratio was about $d_c/L_c$ - 1/1600–1/1800. When using hard X-ray pulses (modern XFELs have ranges up to 25 keV), the attenuation length can be several units to tens of mm in various materials. The ability to change the attenuation length by varying the energy of photons of the XFEL pulse opens up fundamentally new prospects for the controlled processing of materials and the creation of cavities of the required topology. With the current abilities of the available X-ray lasers at EuXFEL and LCLS-II, generating ultrashort high-energy X-ray pulses with up to MHz repetition rate, this work laid a solid background for producing up to a million nanochannels per second. This is the quintessence of the presented experimental evidence, supported by thorough computational analysis, that a single pulse, in fact, can be produced by a single pulse

generated by a MHz repetition rate laser. It opens a new avenue for the development of nanochannel sieves for lab-on-chip applications in any material, including those that are not transparent to optical lasers.

It is worth noting that structures in the form of extended cavities should form when irradiated with X-ray photons with energies of several keV, not only in LiF but also in ceramics, semiconductors, and metals. As an example, it was already demonstrated in work[52] that a similar structure in the form of an extended cavity with a diameter of 4 μm (aspect ratio 1:10) was produced in the Si semiconductor after irradiation with a focused XFEL pulse ($d_{FWHM} = 1$ μm, $\tau = 20$ fs) with a photon energy of 10 keV and an absorbed energy density $\xi = 430$ kJ·cm$^{-3}$ per pulse. The damage thresholds of Si and LiF are similar (5.82 kJ·cm$^{-3}$ per pulse and 4 kJ·cm$^{-3}$ per pulse), and the densities of these materials are close (2.65 g·cm$^{-3}$ and 2.64 g·cm$^{-3}$). The authors of ref. 52 assumed that this structure was formed by the ejection of matter from the channel along the propagation axis of the XFEL beam. However, our work reveals the mechanism of the formation of such long cylindrical cavities in solid materials under the influence of a hard X-ray pulse. Radially expanding shock wave and the following rarefaction wave are responsible for the compression of the surrounding material and formation of the void–so-called ("radial extrusion"). We should also underline that the unloading wave creates a density rarefaction in the center; see the diagrams in Fig. 5a1. However, this region remains filled with a hot medium (first plasma, then supercritical gas or fluid). The temperature and density of this fluid and the radius of the cavity are determined by the energy density supplied. If this energy is high, the substances are quickly removed, and a hot gas with a low density remains in the channel. The isochoric cooling eventually creates conditions for the formation of condensate droplets (or perhaps snowflakes) in this gas, which adhere to the cold walls of the channel. If this energy is not too high, the density of the hot liquid after the waves subside can be considerable (higher than the critical density), and gas bubbles begin to form (boiling) in the isochoric process of slow cooling. If the fluid is sufficiently dense and cooling is rapid, the fluid may rupture. Our MD calculation illustrates the latter process of cavitation. The mechanisms indicated for the formation of the final cavity, therefore, depend on the energy invested, but in both mechanisms, a rarefaction wave acts in the initial phase. We should point out that the described mechanisms apply to any type of material where a high-energy-density region is rapidly created. In this region, distinctions between material properties become insignificant, since a very high-pressure plasma forms. The surrounding cold material then moves like a fluid under this pressure. This also indicates the universality of the hollow channel formation mechanism along almost its entire length, with the exception of the channel's very end (zone 3). At this point, the material's strength and thermophysical properties might lead to differences in how various materials behave. Therefore, similar hollow structures should be observed when irradiating metals with XFELs, for which experimental results are currently lacking.

Another important area is, of course, the investigation of polymorphic (allotropic) transformations in the solid phase. Ultrashort optical laser-induced microexplosion in confined geometry has already demonstrated the formation of previously unknown high-pressure material phases such as body-centered aluminum (bcc-Al)[53] and two new energetically competitive tetragonal structures of silicon[20]. These were the so-called st12-Si, an analogous structure to the well-known st12 phase of germanium, and an experimentally new bt8-Si structure, both were identified in the laser-modified volume by electron diffraction and complementary ab initio random structure searches[21]. In addition, electron diffraction also revealed the presence of other yet unidentified Si phases. Other examples include switching the valence in Fe-atoms in olivine[54], the spatial separation of ions with different mass, and the formation of molecular oxygen inside the voids in oxides[19], and the formation of N-vacancies in c-BN crystals[22]. All these transformations occurred in near-spherical-shaped

nanovolumes contained within a thin surface layer. Ultrashort XFEL pulses in hard, several keV, X-ray energy open up completely unique possibilities for highly targeted laser restructuring of materials. The presented results constitute a robust benchmark for the formation of a broad range of exotic high-pressure materials with extraordinary properties and for studies of highly non-equilibrium electron and ion dynamics in warm dense matter.

In summary, the presented results offer new insights into the utilization of high-brilliance X-ray pulses to create arrays of high aspect ratio nanochannels in any solid material for biosensing applications, as well as for fundamentally new fluidic phenomena observed in nano-fluidic devices. Future prospects undoubtedly involve extending experimental results using XFEL beams on various materials, including non-transparent materials in the optical spectral range, and the investigation and control of created nanocavity morphology as a function of irradiation parameters and photon energy.

## Methods

### Target details and conditions of exposure

A circle LiF crystal of 2 mm thickness and 20 mm diameter (two-sided polished with $R_z = 0.05$ μm) was used as the target sample. To ensure consistency and reproducibility, each exposure was performed on a fresh sample location.

We investigated two single-pulse irradiation regimes of a LiF crystal with pulse energies $E_1 = 26.7$ μJ per pulse and $E_2 = 81$ μJ per pulse (10% error). In both regimes, the sample surface was at the point of best focus. The reduction in pulse energy was achieved by using a beam attenuator. To estimate the extent of the impact on the sample, we used the absorbed energy density per pulse $\xi$ on the LiF surface, calculated as:

$$\xi[z = 0] = \frac{E_{pulse}}{S * d_{att}} = \frac{E_{pulse} * 4 * \ln(2)}{\pi * d_{att} * d_{FWHM}^2} \quad (1)$$

where $E_{pulse}$- pulse energy, $d_{att}$- attenuation length in matter, $d_{FWHM}$- beam size at half maximum.

Considering the beam size and the attenuation length $d_{att} = 475$ μm of 9 keV photons in LiF[55], the absorbed energy density was $\xi_1 = 297$ kJ·cm$^{-3}$ and $\xi_2 = 895$ kJ·cm$^{-3}$ on the sample surface. The attenuation of the beam as it propagates in LiF is due to two factors. The first is the absorption in the substance, and the second is the attenuation due to the geometric divergence of the beam. Asymptotically, outside the waist zone, the beam is not a cylinder due to the broadening, but a cone with a very small but finite opening angle at the tip of the cone. The density of the absorbed energy $\xi$ on the beam axis $z$ is the same:

$$\xi[z] = \xi_0[z = 0] \exp(-z/d_{att}) \frac{S_0}{S(z)} \quad (2)$$

here $\xi_0$ –density of absorbed energy on the target surface, i.e., at $z = 0$, $S_0$ and $S(z)$ –cross-sectional area of the beam at the surface and depth $z$ of the crystal, respectively.

The radius $r_{beam}(z)$ of the beam cross-section at depth $z$ is equal to:

$$r_{beam}^2(z) = r_0^2 \left[ 1 + \left( \frac{z}{z_R} \right)^2 \right] = r_0^2 \left[ 1 + \left( \frac{\ln(2) z \lambda M^2}{2\pi r_0^2} \right)^2 \right] \quad (3)$$

where $r_0$ is the spot radius of the beam at the focal plane, $z_R$ is the Rayleigh range, $\lambda$ is the wavelength of the photon, and $M^2$ is the beam quality factor.

Using Eqs. (2 and 3) and the parameter $M^2 = 3$, which we found earlier in work[35] for a given focusing of the XFEL beam, we plotted the

dependence of the decay $\xi[z]$ in the LiF crystal, presented in Fig. 3c:

$$\xi[z] = \frac{\xi_0[z=0]}{1 + \left(\frac{\ln(2)z\lambda M^2}{2\pi r_0^2}\right)^2} \exp(-z/d_{att}) \tag{4}$$

## Investigation of cavity morphology

After irradiation, the LiF crystal was examined using various readout systems, see Fig. 1b: Step I–a SEM was used to examine surface damage – HELIOS NANOLAB 600I, FEI. Step II–to obtain information about the intensity distribution of the incident beam inside the crystal, a Carl Zeiss LSM 700 confocal laser scanning microscope in fluorescence mode was used. Since the LiF crystal is a fluorescent medium, X-ray irradiation creates CCs in it, the distribution of which corresponds to the irradiation area[35,56]. The measurements were performed using an excitation laser with a wavelength of $\lambda_{las} = 488$ nm and an objective with a magnification of ×100. The signal was recorded layer by layer with a layer thickness of $\Delta z = 1\,\mu m$ at a spatial resolution of 0.25 μm (field of view 200 × 200 μm²). Step III–For several test points, the FIB-SEM, FEI Helios NanoLab 600I) technology was used, which made it possible to examine the structure of the destruction area in its cross-section, i.e., its depth. In contrast to an electron microscope, the FIB inherently destroys the sample. When the high-energy ions hit the sample, they blast atoms off the surface. The FIB tool was used to etch the LiF sample layer by layer in the area next to the damage, while an SEM was used to visualize the morphology of the damage (perpendicular to the XFEL beam propagation plane). The preparation of the sample before milling consisted of a thin gold coating (a few nm) on the whole surface, followed by a thin protective layer of platinum locally deposited on the top of the hole. The platinum layer can be seen in the cross-section SEM images at the hole entrance. Then, FIB milling was performed, varying the current from nA (for opening a viewing section) to pA (for precise slicing of the hole).

## Simulations

We used a simulation approach of simulations to characterize our channel formation. Three types of simulations were used: smooth particle hydrodynamic (SPH), MD, and HD. The results of the last are shown in Supplementary Information.

Firstly, our 2D SPH code equipped with the LiF damage model[35] (see Supplementary Information II) was used to solve the problem of cavity expansion in the zones 1 and 2 regions (see Fig. 3a). Since the heating zone of the incident XFEL beam in the material acts on the cold solid material like a gas piston at pressure changing with time, the problem can be simplified by replacing the hot material zone with a boundary condition for the pressure $P(t)$ set at the boundary between hot and cold material. In SPH simulations, to account for the initial laser heating within a cylinder with a radius $r_{beam} = 205$ nm, a uniform heating is given by the specific energy $e(z) = \xi(z)/\rho$, calculated according to Eq. (4) at a certain depth $z$ from the sample surface.

The instantaneous heating due to the absorption of a laser pulse is accompanied by a pressure jump of $\Delta P(z) = (\gamma - 1)\rho_0 e(z)$, where $\gamma \simeq 1.75$ is the adiabatic exponent, $\rho_0 = 2.65$ g·cm⁻³ is the nominal density of LiF. Such localized heating leads to the formation of a strong shock wave, which propagates through the "cold" material (outside the heating area) and leads to its damage. However, it is worth noting that the amplitude of the shock wave decreases quite rapidly with moving away from the heating area, which limits the damage area to about ten microns or less, see insets (f2, f3) to Fig. 4.

To analyze the dynamics of the cavity opening near the surface (zone 1 in Fig. 3a), the modeling was performed in a 3-dimensional environment, for which the axial symmetry of the flow was not used. Thus, although the near-surface flow has two spatial coordinates $(r, z)$, the simulation uses SPH particles that are initially arranged in a grid in

the Cartesian $(x, y, z)$ coordinate system–accordingly, this is referred to as a 3D simulation. Heating of material was applied instantaneously by increasing the internal energy of SPH particles according to a given radial Gaussian distribution of energy deposition within a cylinder, i.e., the gas piston model was not used in those 3D simulations.

MD simulations of LiF were performed with a newly developed pairwise potential for interaction between Li+ and F- ions in the following form:

$$V_{ij}(r) = \left[q_i q_j \frac{\exp(-r/d)}{r} + \frac{a_4}{r^4} + \frac{a_6}{r^6} + \frac{a_8}{r^8}\right] f(r, r_c) \tag{5}$$

where $q_i$ and $q_j$ are the known electrical charges of $i$- and $j$-ions, while $d, a_4, a_6, a_8$ are fitting parameters presented with more detailed description in Supplementary Information III. The shielding with length $d$ is used to escape time-consuming calculations of the long-range Coulomb forces.

The smoothing function $f(r, r_c)$ causes the above potential to go to zero with an interatomic distance $r$ approaching the cutoff radius $r_c = 0.875$ nm. The potential parameters were fitted to the known cold pressure curve[57,58] in a wide range of compression and stretching by using the stress-matching method[59]. The obtained potential provides the melting temperature of 990 K and reproduces the experimental difference of 25% between the densities of molten and solid LiF at the melting point. All MD simulations were performed with our in-house parallel code[60].

## Data availability

All data are available in the main text or Supplementary Information. Source data are provided with this paper. The data that support the findings of this study are available from the corresponding authors upon request. Data recorded for the experiment at the European XFEL are available at https://doi.org/10.22003/XFEL.EU-DATA-002575-00. Source data are provided with this paper.

## Code availability

SPH simulations were performed with our in-house private code CSPH-VD³ based on the contact SPH algorithm with the Riemann solver[47] and utilizing the Voronoi dynamical domain decomposition (VD³) for parallelization[48]. MD simulations were performed with our in-house code MD-VD³ based on the classical MD method with the velocity Verlet integrator and utilizing the VD³ [60]. The code is available upon request to 6asi1z@gmail.com. The HD code is based on the 2D Baer-Nunziato model of multifluid hydrodynamics[61] using an HLLC-type Riemann solver. The code is available upon request to petchu@mail.ru.

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

## Acknowledgements

The experimental results presented in this paper were obtained before February 2022. We greatly thank Alexander Pelka, Hauke Hoeppner, Carsten Baehtz, Cornelius Strohm, Christian David, Clemens Prescher, Seniutinas Gediminas, Patrik Vagovic, Sebastian Goede, and Andreas Schropp for their valuable contribution and support during the experiment. We acknowledge European XFEL in Schenefeld, Germany, for the provision of X-ray FEL beam time at HED SASE2 and would like to thank the staff for their assistance. T.A.P. acknowledges financial support provided by JSPS KAKENHI Grant numbers 21K03499, 24K06988. L.J., J.Ch., V.V., V.H., and T.B. acknowledge financial support provided by the Czech Ministry of Education, Youth and Sports (grant no. LM2023068). This work used the ACT node of the NCRIS-enabled Australian National Fabrication Facility (ANFF).

## Author contributions

Conceptualization: S.S.M., S.A.P., and U.Z. Methodology: S.S.M., S.A.P., and T.A.P. Conducting an experiment: S.S.M., M.M., M.N., T.A.P., K.A., Z.K., V.C., E.B., J.P.S., I.M., V.V., V.H., T.B., J.C., L.J., U.Z., and S.A.P. Investigation: S.S.M., V.V.Z., S.Yu.G., T.A.P., N.A.I., V.A.K., and A.V.R. Visualization: S.S.M., V.V.Z., S.Yu.G., P.C., and N.O. Readout procedure: S.S.M., T.A.P., T.S., A.T., L.R., A.V.R., and S.J. Modeling: S.Yu.G., V.V.Z., N.A.I., V.A.K., Yu.V.P., V.S., P.C., and E.A.P. Writing—original draft: S.S.M. Writing—review and editing: T.A.P., V.V.Z., N.A.I., S.A.P., L.J., V.H., A.V.R., L.R., and R.K.

## Competing interests

The authors declare no competing interests.

## Additional information

[1]Joint Institute for High Temperatures of Russian Academy of Sciences, Moscow, Russia. [2]Institute for Computer Aided Design, Russian Academy of Sciences, Moscow, Russia. [3]Institute for Open and Transdisciplinary Research Initiatives, Osaka University, Suita, Japan. [4]Landau Institute for Theoretical Physics of Russian Academy of Sciences, Chernogolovka, Russia. [5]The facility at Material Science Research Center, Japan Atomic Energy Agency, Sayo, Japan. [6]Laser Physics Centre, Department of Quantum Science and Technology, Research School of Physics, Australian National University, Canberra, ACT, Australia. [7]Optical Sciences Centre and ARC Training Centre in Surface Engineering for Advanced Materials (SEAM), School of Science, Swinburne University of Technology, Hawthorn, VIC, Australia. [8]Tokyo Tech World Research Hub Initiative (WRHI), School of Materials and Chemical Technology, Tokyo Institute of Technology, Tokyo, Japan. [9]European XFEL, Schenefeld, Germany. [10]Università degli Studi di Milano Bicocca, Milano, Italy. [11]Department of Radiation and Chemical Physics, Institute of Physics, Czech Academy of Sciences, Prague 8, Czech Republic. [12]Graduate School of Engineering, Osaka University, Suita, Japan. [13]Institute of Laser Engineering, Osaka University, Suita, Japan. [14]HB11 Energy Holdings, Freshwater, Sydney, NSW, Australia. ✉e-mail: seomakarov28@gmail.com

