## [Transparent Peer Review file redacted · Nature Communications]

Formation of high-aspect-ratio nanocavity in LiF crystal using a femtosecond X-ray free-electron laser pulse

Corresponding Author: Dr Sergey Makarov

Version 0:

Reviewer comments:

Reviewer #1

(Remarks to the Author)

Reviewer report on the manuscript entitled "Formation of high-aspect-ratio nanocavity in LiF crystal using a femtosecond of x-ray FEL pulse" by Sergey S. Makarov et al.

:: General comments

The authors present a very detailed and comprehensive study of single-shot x-ray FEL induced nanocavities in a LiF crystal. The post-mortem analysis of the sample, carried out using laboratory techniques (SEM, FIB-SEM, LSM, etc.), is corroborated by numerical simulations. Overall, the presented work is commendable.

As a general impression, I question the use of an outstanding FEL source like EuXFEL, which can operate at MHz (!) repetition rate, for single-shot damage on materials. Alternative low repetition rate x-ray FELs are probably more appropriate for this class of experiments.

The authors do not seem interested in exploiting the real potential of their FEL source and do not mention the possibility of exploring the picosecond and nanosecond formation dynamics of the FEL-induced nanocavity. Some kind of pump-probe approach, using a fs-laser in combination with the intense FEL pulse or even a double x-ray pulse, would make their study more intriguing.

The authors emphasize the importance of demonstrating the formation of a deep cylindrical submicron cavity. In particular, they claim that the "radial extrusion" process is the main mechanism behind the cavity formation. This conclusion may attract the attention of some specialized readers but probably does not represent significant information for the broader FEL community. More interestingly, the authors suggest the possibility of investigating polymorphic FEL-driven phase transitions in solids. However, they do not explain if there exist specific experimental approaches that enable monitoring the formation of sample allotropes around the nanocavity on the picosecond and nanosecond timescales after FEL exposure.

:: Detailed Comments

Line 70: "required MJ/cm³ level of energy concentration" should be "required MJ/cm³ level of energy density."

Line 77: "XUV/x-ray FELs represent promising tools for direct nano-patterning of solids." X-ray FELs can certainly be used for nano-patterning; however, the authors should clarify that the operational costs and limited beamtime of FELs do not permit considering them ideal facilities

for regular operation of these kinds of applications.

Line 173: There is a strike-through in the number "2".

Line 227: There are irregular fonts here

:: Conclusions

The employed methods are valid, and the quality of the presentation is very good; however, the significance of the conclusions is rather limited, especially considering the actual potential of the EuXFEL source. Optimization and constant improvements of x-ray FELs achieved in the last decade foster the use of these sources for more ambitious experiments, such as those involving ultrafast or nonlinear phenomena.

Instead, the key result of the presented study is limited to the characterization of FEL nanocavities in LiF using conventional laboratory instrumentation and theoretical simulations. This kind of investigation was innovative 15 years ago when x-ray FELs became available, but today its impact has decreased and does not justify publication in Nature Communications, in my opinion. I would recommend publishing the manuscript in an alternative specialized journal.

Reviewer #2

(Remarks to the Author)

In this study, holes with an aspect ratio of more than 1:1000 are drilled using single-pulsed hard X-rays. The diameter and the depth of the obtained holes are measured by a combination of fluorescence LSM and FIB+SEM. The authors try to explain the formation of the holes including the observed three substructures by the three different simulation methods: SPH, MD, and HD.

The determined huge aspect ratio of the holes is exciting and processing holes with a diameter from 1 – 2 μm and a depth of more than 1 mm using single-pulsed radiation is unique to the best of my knowledge. The obtained experimental results may contribute to different promising applications as discussed by the authors.

Despite the unique experimental results, I have to mention a few critical points which should be addressed before publication.

Introduction:

The authors discuss conventional drilling using laser radiation. The description is too general, as different drilling strategies and techniques exist enabling also a high aspect ratio of the holes.

The section should be checked in terms of the references being often missing to some statements, such as "However, large length-to-diameter ratios cannot be achieved in this way (as a rule, this ratio does not exceed values in the order of one or few tens)."

"This situation differs fundamentally from the usual picture of laser ablation with the formation of a shallow crater whose depth is small compared to the beam diameter, see e.g.1,19." The authors should discuss why the ablation mechanism differs fundamentally and also what are the differences to the creation of voids inside of transparent materials using highly intensive laser radiation.

Morphology of the generated cavity in combination with the simulations:

The sections should be revised as some explanations and descriptions of simulated processes for forming the cavity are often only given in the text and are not supported by the figures of simulations. Either the authors have to extend the manuscript, or I recommend adding it to the supplementary material.

Morphology of generated cavity:

"To verify that the observed absence of the PL signal in the LiF images in the central region of the XFEL beam in Fig. 3A is due to the absence of matter (cavity), we cut the LiF sample layer by layer near the expected cavity to a depth of a few tens of micrometers from the surface (see Methods for details)" According to this sentence, the LSM measurements are just verified up to a few tens on micrometers, but in Fig 4 D the maximum measured depth is more than 700 μm . Which one is correct? Additionally, the final depth of the drilling hole is not verified. The authors also should add some SEM micrographs at positions several 100 μm deep in the material. Those micrographs might be added to the supplementary material. If the final depth of the holes is more than 1 mm, shouldn't they are visible in an optical microscope in transmission through the side?

Multi-method simulations:

The authors should also discuss the absorption process as due to the excitation of the material a decrease in the optical penetration depth might be expected, as for usual ultrafast laser radiation in the VIS range.

The authors present three different simulation methods to explain different processes in the formation of the final hole. The authors should discuss the limits of each simulation method to clarify why each method can explain only the selected use case. From my point of view, MD represents the most powerful method, and as a simulation of the complete hole is not a limitation, so why not only MD simulations are used?

Discussion:

Most paragraphs do not discuss the results but provide instead an outlook for further investigation or a comparison with other researchers. Therefore, the section should be revised to represent a discussion of the results.

Reviewer #3

(Remarks to the Author)

NCOMMS-24-27119 "Formation of high-aspect-ratio nanocavity in LiF crystal using a femtosecond of x-ray FEL pulse" by Makarov et al.

Makarov et al. propose a method of volumetric ablation and nanostructuring of materials using ultrashort (20 fs) hard x-ray pulses at 9 keV provided by the FLASH source in Hamburg. The focused x-ray pulse can generate high aspect ratio nanostructures in the volume of a dielectric LiF sample. A range of theoretical continuum and atomistic models are used to simulate the formation of the cavity and the surrounding pressure effects. Claim is made that the technique can be used to nanostructure bulk metallic materials where no alternatives for volume ablation exist. The modification process is based mainly on dimensional analysis and electron microscopy, and supported by theoretical models is associated with shock induced amorphization and cavitation.

Altogether this is a very solid scientific work, where the interaction of short x-ray pulses for nanoprocessing is becoming highly relevant. Demonstrating their importance for a range of structures in materials is appealing and interesting. The discussion is pertinent and insightful, with sometimes a dispersion in the analysis coming from the different models that obscures a clear conclusion on the nanocavitation process and the surrounding amorphization.

Nevertheless, some of the claims leave the impression of being rather speculative or at least far-fetched. I will indicate below some

1. The manuscript is scarce in details concerning matter excitation or ionization. It is expectable that saturation of x-ray absorption in the FEL micro-spot occurs, given the high dose. No further details are given about the ionization mechanisms, collisional processes, inner-shell decay or scattering, and how these affect the energy deposition profile. No information is given on how fast energy couples to the material at the electronic and molecular level, and how fast phase transitions may occur. I feel that such a discussion may be important to state the level of energy density as the main ansatz for the simulation.
2. The attenuation depth (at 1/e) of 475 μm and the existence of a cavity at more than 1000 μm in depth may suggest that at surface the energy density is about one order of magnitude above the cavitation threshold. Below the cavity end one notices (Fig 3) modification that is not pressure related but merely a phase transformation. Is the possible saturation expected to give a different energy density profile with less peak energy density and larger widths as reported here based solely on geometrical consideration? A study of the threshold of the cavity process with different energies down to the threshold (and not in a single pulse) could illuminate the energy evaluation (according to the estimation a cavity should be formed for input energies in the 1 μJ range, four times above the modification threshold given in Ref. 25). I note also the remarkable stability of the cavity along propagation (1000 μm) despite the variation of the energy density for more than an order of magnitude (Fig. 3A).
A discussion based on the energy density required to kick-start a process is given at page 9. The main argument is dimensional, saying that melting or cracking occur in regions where estimated local energy density based on pulse profile is too small so the process is not driven by local energy but by shock pressure coming from the central area, triggered by fast isochoric heating (the argument is that the melting area has a radius of 0.6 μm with respect to the 0.3 μm at 1/e² expected from the x-ray pulse, low energy case). A full proof of such discussion would require the measured metrology of the X-ray pulse at the measured plane as scattering and absorption saturation may locally modify the profile with respect to Fig. 2C. Secondly, it is important to highlight the process dynamics; for example, a relaxation of a lower density amorphous phase upon cooling may create (due to the density mismatch) type of stress components that will generate spallation or cracking of the adjacent areas. This will provide a hint to the state of the material when it suffers the modification, as its evolution may depend on it. Third, thermal transport may be active as long as the sample is hot and heat may propagate in the surroundings. More arguments may be needed to support the statement in addition to the crude geometrical evaluation.
3. The suggestions several times in the text of TPa pressure levels as well as polymorphic transformations are not really supported here, though these appear as key elements of the discussion.
4. The authors suggest, quite understandably, a strong interest for metals however their use case is a dielectric where optical Bessel Gauss pulses can produce similar high aspect cylindrical geometries for nanocavities that do not require MJ/cm³. This puts forward some inconsistencies in the interpretation of nanocavity formation with optical and x-ray pulses, though dimensionally cavities and surrounding amorphous or stressed regions are similar. The dynamic behaviour suggested here seem equally different from the one reported with optical pulses and this issue should be addressed. Phase transitions, mismatch in amorphous and crystalline densities, and heat transport were also suggested to explain modifications observed post mortem, without invoking high levels of mechanical shock.
5. A comprehensive and relevant simulation package is used to analyse the hydrodynamics and MD aspects of the process starting with the evaluated input energy density profile (hence the need to accurately confirm it). Timescales in the range of ns are found (Fig. 4C) while sub-ns times are given in Juodkazis et al. PRL 2006 and several tens of ns are given in

(Nguyen et al. Ultrafast Sci. 2024). Such discrepancies may perhaps be explained by the nature of the material and irradiation conditions but I feel correlations should be made as similar processes are discussed. What would be the theoretical energy threshold for cavity formation when all heating, structural transformation, and relaxation processes are accounted for and how the phase transformation of the material influences the process? Some elements of the discussion are given at Fig. 6 but the times of several ps shows apparently some inconsistencies with Fig. 3. The MD simulation brings into the discussion the cooling of the hot melt and its influence on the cavitation, while so far the discussion was dominated by shock launch. Perhaps a conclusive and convergent discussion coming from the different codes used here may put forward the main common elements, with a focus on the nanocavity formation; the key element of interest.

6. To the stress distribution, there may be several elements leading to such a distribution. As a general note, radial transport will create perpendicular stress components and similar polarization patterns.

In conclusion, this is a very interesting and original report. It reads nicely even though sometimes the discussion gains in complexity when surface and volume processes are discussed and mixed. Nevertheless, I feel that at this point, despite the high level of scientific discussion, the comprehensive analyses and theoretical simulations, and the originality of the irradiation conditions, the manuscript does not illuminate new physical aspects related to the interaction of short energetic pulses and matter in the cavitation range. Cavities and nanobubbles were reported in dielectric materials in several occasions and their conditions analysed, with perhaps conclusions that may also be relevant to this work and also alternative formation scenarios. The cylindrical relaxation geometry was found optimal for generating high aspect ratio nanochannels up to 1:10000. What appears notable here is the high energy density values, that may potentially create specific relaxation conditions for the X-ray pulse. Their validation is therefore critical to the scenarios proposed here and to the conclusive section of the manuscript. The claimed extreme conditions need therefore further experimental insight to be confirmed (with arguments in addition to the dimensional analysis using a pulse profile propagation in "vacuum"), as similar post-mortem damage and cavity appearances can be achieved optically at lower energies (energy density at least one order of magnitude lower), with similar sections for cavities and surrounding amorphous regions. The authors should demonstrate why secondary processes such as scattering and absorption saturation as well as fast diffusion cannot increase the interaction section. The similarities and differences with respect to the cavitation induced by optical (Bessel) beams either in terms of energy density or process dynamics should be outlined. At the same time elements of beam metrology in interaction at high doses should be given in support of the dimensional analysis. This will help to clearly indicate the shock scenario as most probable, opposite to a transformation related to local energy density and not driven by propagating mechanical waves. References to extreme polymorphic transformations should be demonstrated otherwise they appear speculative in the context.

Therefore, despite the interest of the report, I cannot give a positive recommendation for publication in Nature Communications. However, I may reconsider if direct and convincing experimental proof of high shock levels is given together with the associated polymorphic transformations, transient or permanent, including the validation of the energy density levels.

Miscellaneous.

The addition of drawings in Fig.2 makes it difficult to evaluate what the authors correlate with melting. The SEM appearance is not by itself the strongest argument for melting.

Version 1:

Reviewer comments:

Reviewer #1

(Remarks to the Author)

The authors have thoroughly revised the manuscript, addressing most of the comments and criticisms raised by the reviewers. They have done a commendable job, resulting in a significantly improved version. The novelty of the experiments, as well as their potential impact on applications, is now clearer. The message conveyed by the manuscript is more effective. In its current form, the work can be considered for publication in Nature Communications.

Reviewer #2

(Remarks to the Author)

The authors answered my remarks and updated the manuscript accordingly. The manuscript is now much clearer and easier to understand. It can be accepted in terms of content.

Reviewer #3

(Remarks to the Author)

The authors gave new insights in their detailed response. I emphasize very interesting physical discussions, tackling a wide range of assumed conditions. I note at the same time a technologically-gearred introduction which seems far-fetched for XFEL as similar performances can be reached easier with optical beams. Moreover, resemblances of X ray damage with optical damage on surface (exfoliation present on ionic crystals in fs laser ablation experiments) and in the bulk (cavities are expected for tight-focused excitation in cylindrical and spherical geometries) raises questions about the expected extreme behaviors. Are extraordinary processes to be expected at 0.3-0.9MJ/cm³ levels given that sub-micron cavities can be obtained with a tenth of deposited energy? Without clear indications (for example the absence of experimental confirmation of sub-ns shock and cavity dynamics), the physics here becomes less singular. Theories and models about materials under constraints were discussed in various occasions. The demonstration I suggested is the structural proof of matter following extreme evolution with a potential dynamic view of the process. Despite the compelling comments and numerical methods, such a proof was not given and statements remain partly speculative. The novelty remains utilization of XFEL source (notwithstanding the proof-of-concept material also transparent at optical wavelengths). I have the feeling that strength lies more in the perspectives that were proposed. I therefore maintain the initial recommendation.

Version 2:

Reviewer comments:

Reviewer #3

(Remarks to the Author)

The main question in my opinion is if the manuscript, notwithstanding its quality of the discussion -which is very good and particularly rich-, demonstrates physical transformations not achievable by other means or illuminate unique mechanisms of transformation. These points remain unanswered to me.

-Choice of material: I agree with the authors with the limitation of optical beams for non-transparent materials. If they work remarkably for generating nanostructures with high aspect ratio in dielectrics (and also with minimal nonlinear distortions, the case of non-diffractive beams) they cannot work on other materials. Therefore, my interrogation, if this is a key point, why not demonstrating the technique on materials that have no other alternatives? Many characterization techniques exist nowadays to probe the development of the nanochannel in the bulk of non-transparent materials. The argument of the first use of a XFEL pulse to this end should be accompanied by a demonstration that actually illuminates this advantage.

-To the mechanism and the discussion of extreme mechanics involved in the void appearance. The formation of volume nanovoids with optical beams (including high aspect voids formed by Bessel beams -and I'm referring to volume structures with aspect ratios exceeding 100 or 1000 and not to shallow surface craters) does not necessarily involve shock compression, according to recent studies. The authors presented (quite selectively in my opinion) references to this extent as also occurring in the case of optical beams, but they would agree with me that the image of residual traveling elastic waves is not a proof for shock compression (Yu et al <https://doi.org/10.1364/OE.26.021960>). Alternative scenarios exist for high aspect nanostructuring occurring at far lower energy density (up to hundred times lower). The proposed scenario for X-ray is plausible, possibly probable at high energy density, but not every formation of a nanocavity should be seen through the prism of extreme physics. If high energy physics transformation is implied, experimental evidence is due. Structural characterization in the material neighboring the cavity illustrating massive distortions, density changes or new high-pressure structural arrangements, could have provided strong evidence to the proposed scenario.

To conclude, this is a solid and enlightening piece of physical discussion but it insufficiently illuminates the process of nanovoid formation and its unique manifestation under X-ray exposure, with experimental evidence being in my view necessary to validate the hypotheses. One could perhaps go with the argument of first demonstration of linear absorption volume structuring, but the mechanism validation is important for the community after years of employment of microexplosion concepts. The points indicated in the perspectives will nonetheless establish ultrafast X-ray irradiation as unique nanostructuring means.

Version 3:

Reviewer comments:

Reviewer #1

(Remarks to the Author)

I really appreciate the considerable effort made by the authors to clarify all the points raised by the reviewers. The presented work is highly commendable, resulting from an impressive combination of experimental and theoretical research. While the findings may call for further progress and refinement of the adopted methodologies, this is quite natural in frontier research with x-ray FELs. At the current stage, the results reported by the authors already have considerable impact and are likely to stimulate the scientific community to further explore the field. Possible next steps and improvements could be briefly outlined in a few short perspective sentences at the end of the Discussion. Overall, I confirm my opinion that the manuscript fully deserves publication in Nature Communications.

Reviewer #2

(Remarks to the Author)

As already stated in my last decision, the manuscript can be accepted in terms of content. The newly added explanations further enhance understanding. Further experimental investigations are not necessary for this study, in my opinion.

We would like to thank the referees for the work they performed and their impression of the manuscript.

To make proofreading easier, we have highlighted in **red** all differences with the previous versions. The following document contains our answers (written in **blue**) to the referee's comments (written in **black**).

After taking into account the reviewers' criticism regarding the importance and possible applications of the results of our work, we completely rewrote the sections abstract, introduction and discussion to cover these aspects in more detail.

On behalf of all co-authors,

S. Makarov

REVIEWER COMMENTS

Reviewer #1 (Remarks to the Author):

Reviewer report on the manuscript entitled "Formation of high-aspect-ratio nanocavity in LiF crystal using a femtosecond of x-ray FEL pulse" by Sergey S. Makarov et al.

General comments

- 1. The authors present a very detailed and comprehensive study of single-shot x-ray FEL induced nanocavities in a LiF crystal. The post-mortem analysis of the sample, carried out using laboratory techniques (SEM, FIB-SEM, LSM, etc.), is corroborated by numerical simulations. Overall, the presented work is commendable.**

As a general impression, I question the use of an outstanding FEL source like EuXFEL, which can operate at MHz (!) repetition rate, for single-shot damage on materials. Alternative low repetition rate x-ray FELs are probably more appropriate for this class of experiments.

Reply: Today only XFELs such as EuXFEL, LCLS and SACLA XFEL allow to perform experiments for investigation long nano-cavities in solids. This is possible thanks to a wide range of photon energy (several keV are required), high intensity and pulse energy (~ 1 mJ) of such X-ray sources. EuXFEL as well as LCLS and SACLA XFEL are often used in single-pulse mode for a variety of physics applications, especially for high energy density physics experiments (MHz mode is rather intended for biological/chemical research).

This manuscript presents the very first demonstration of the ability to form a nanochannel in solid material with a single X-ray pulse generated by an XFEL. With the current abilities of the available X-ray lasers at EuXFEL and LCLS-II generating ultrashort high-energy X-ray pulses with up to MHz repetition rate this work laid a solid background for producing up to a million of nanochannels per second (!). This is the quintessence of the presented experimental evidence, supported by thorough computational analysis, that a nanochannel in fact can be formed by a single pulse generated by a MHz repetition rate laser.

2. **The authors do not seem interested in exploiting the real potential of their FEL source and do not mention the possibility of exploring the picosecond and nanosecond formation dynamics of the FEL-induced nanocavity. Some kind of pump-probe approach, using a fs-laser in combination with the intense FEL pulse or even a double x-ray pulse, would make their study more intriguing.**

Reply: The formation of nanochannel with more than $\times 1000$ times aspect length-to-diameter ratio with a single XFEL pulse is a first step on the long way toward the understanding the mechanism of high energy density deposition above the strength of material inside the bulk, the following generation of radially-symmetric shockwave, and analysis of the ‘frozen’ compressed material states surrounding the void formed in the isochoric conditions of conservation of mass.

The essence of using ultrashort laser pulses with sub-picosecond pulse duration is to create the high energy density required for transition of the materials to the state of warm dense matter, which in turn leads to a strong shock wave and the following rarefaction wave responsible for the void formation. The picosecond-range and nanosecond pulses are longer than the electron-ion energy transfer time and thus are not suitable, as suggested by the Reviewer, for nanochannel formation. According to our estimates the electron-ion energy exchange time in LiF is ~ 4.8 ps [*Phys. Rev. B* 73, 214101 (2006)]. The energy deposition time should be a few times shorter, < 1 ps, to create the high-energy density state of matter before the deposited laser energy could dissipate into the bulk through the electron-ion collisions.

We thank the Reviewer for the advice of using pump-probe experiments to observe the process of material restructuring under the shock wave propagation. The pump-probe approach to observe the shock-wave dynamic and analysis of the surrounded compressed shell is exactly the following step we want to apply for XFEL beam time. These plans are presented our publications (see, for example, Ref. [*Phys. Rev. Lett.* 126, 015703, 2021] in the manuscript and recently published in work [*Phys Rev Research* 6, 023101, 2024]). So far, our applications for beamtimes were unsuccessful due to “... *very high complexity of the proposed experiments to be successful within the frames of available beam time*”.

3. **The authors emphasize the importance of demonstrating the formation of a deep cylindrical submicron cavity. In particular, they claim that the "radial extrusion" process is the main mechanism behind the cavity formation. This conclusion may attract the attention of some specialized readers but probably does not represent significant information for the broader FEL community. More interestingly, the authors suggest the possibility of investigating polymorphic FEL-driven phase transitions in solids. However, they do not explain if there exist specific experimental approaches that enable monitoring the formation of sample allotropes around the nanocavity on the picosecond and nanosecond timescales after FEL exposure.**

Reply: We agree that ‘radial extrusion’ should be better presented in physical terms such as “radially expanding shock wave and the following rarefaction wave responsible for the compression of surrounding material and formation of void – so called (“*radial extrusion*”).”. However, we respectfully disagree with the Reviewer’s statement that “**This conclusion may attract the attention of some specialized readers but probably does not represent significant information for the broader FEL community.**” Laser-induced microexplosion in confined geometry has already demonstrated the ability to form new high-density thermodynamically non-equilibrium material states which cannot be formed by any other means due to the record-high rate of energy deposition and quenching. The examples include but not limited to the formation of bcc-Aluminium (*Nat. Comm.* 2, 455

(2011)); at least two new Silicon phases, (*Nat Comm.* 6, 7555 (2015)) and other material non-equilibrium phase transformations into polymorphs with novel and exotic properties [*Phys. Rev. Lett.* 96, 166101 (2006); *Sci. Rep.* 6, 34286 (2016); *Opt. Mater. Express.* 1, 1150 (2011)] which undoubtedly attract attention from a very broad range of scientific and engineering communities.

We stress here that the presented results demonstrate the first significant step towards expansion laser-induced confined microexplosion into the domain of optically non-transparent materials. We should also underline that it is technically difficult to observe the formation of polymorphs over time. But it is possible to observe formed polymorphs that remain in an equilibrium or metastable state. This is done in the same way as was presented in the work [*Sci. Rep.* 6, 34286 (2016)].

:: Detailed Comments

4. **Line 70: "required MJ/cm³ level of energy concentration" should be "required MJ/cm³ level of energy density."**

Reply: Thank you, corrected

5. **Line 77: "XUV/x-ray FELs represent promising tools for direct nano-patterning of solids." X-ray FELs can certainly be used for nano-patterning; however, the authors should clarify that the operational costs and limited beamtime of FELs do not permit considering them ideal facilities for regular operation of these kinds of applications.**

Reply: We agree with reviewer, definitely, today, this is a project of a scientific interest. We have significantly changed the "discussion" section and, in particular, added the following remark: "...We should underline that this manuscript presents the very first demonstration of the ability to form a nanochannel in solid material with a single X-ray pulse generated by an XFEL. With the current abilities of the available X-ray lasers at EuXFEL and LCLS-II generating ultrashort high-energy X-ray pulses with up to MHz repetition rate this work laid a solid background for generating up to a million of nanochannels per second. This is the quintessence of the presented experimental evidence, supported by thorough computational analysis, that a nanochannel can be produced by a single pulse generated by a MHz repetition rate laser. It opens a new avenue for the development of nanochannel sieves for lab-on-chip applications in any material, including those that are not transparent to optical lasers..."

6. **Line 173: There is a strike-through in the number "2".**

Reply: Thank you, corrected

7. **Line 227: There are irregular fonts here**

Reply: Thank you, corrected

:: Conclusions

The employed methods are valid, and the quality of the presentation is very good; however, the significance of the conclusions is rather limited, especially considering the actual potential of the EuXFEL source. Optimization and constant improvements of x-ray FELs achieved in the last decade foster the use of these sources for more ambitious experiments,

such as those involving ultrafast or nonlinear phenomena. Instead, the key result of the presented study is limited to the characterization of FEL nanocavities in LiF using conventional laboratory instrumentation and theoretical simulations. This kind of investigation was innovative 15 years ago when x-ray FELs became available, but today its impact has decreased and does not justify publication in Nature Communications, in my opinion. I would recommend publishing the manuscript in an alternative specialized journal.

Reply: The significance of the presented results, in our opinion, is underestimated by the Reviewer – see our replies 1-3 to previous comments above. The major limitation in using tightly focused optical Gaussian beam in confined microexplosion is imposed by laser pulse diffraction, which limits the radius of the focal spot and thus the absorption depth where the energy is absorbed. To increase the number of atoms rearranging into the unconventionally structured crystal lattices, the development of a method altering a much larger volume of modified structure is required, specifically a method that deposits energy into the electrons faster than it can be transferred to the ions. Ultrashort XFEL pulses in hard, several keV, X-ray energy open up entirely new and unique possibilities for highly targeted laser restructuring of materials. The presented results constitute a robust benchmark for the formation of broad range of exotic high-pressure material with extraordinary properties and to studies of highly non-equilibrium electron and ion dynamics in warm dense matter.

Thank you for your considerations.

Reviewer #2 (Remarks to the Author):

In this study, holes with an aspect ratio of more than 1:1000 are drilled using single-pulsed hard X-rays. The diameter and the depth of the obtained holes are measured by a combination of fluorescence LSM and FIB+SEM. The authors try to explain the formation of the holes including the observed three substructures by the three different simulation methods: SPH, MD, and HD.

The determined huge aspect ratio of the holes is exciting and processing holes with a diameter from 1 – 2 μm and a depth of more than 1 mm using single-pulsed radiation is unique to the best of my knowledge. The obtained experimental results may contribute to different promising applications as discussed by the authors.

Despite the unique experimental results, I have to mention a few critical points which should be addressed before publication.

Introduction:

- 1. The authors discuss conventional drilling using laser radiation. The description is too general, as different drilling strategies and techniques exist enabling also a high aspect ratio of the holes.**

The section should be checked in terms of the references being often missing to some statements, such as “However, large length-to-diameter ratios cannot be achieved in this way (as a rule, this ratio does not exceed values in the order of one or few tens).”

“This situation differs fundamentally from the usual picture of laser ablation with the formation of a shallow crater whose depth is small compared to the beam diameter, see e.g.1,19.” The authors should discuss why the ablation mechanism differs fundamentally and also what are the differences to the creation of voids inside of transparent materials using highly intensive laser radiation.

Reply: Thank you, the first mentioned sentence was removed. We have substantially rewritten the abstract and introduction sections in order to give the reader a better understanding of the significance of this work.

- 2. Morphology of the generated cavity in combination with the simulations:**

The sections should be revised as some explanations and descriptions of simulated processes for forming the cavity are often only given in the text and are not supported by the figures of simulations. Either the authors have to extend the manuscript, or I recommend adding it to the supplementary material.

Reply: Thank you, we have substantially rewritten the simulation section and added more details.

Morphology of generated cavity:

- 3. “To verify that the observed absence of the PL signal in the LiF images in the central region of the XFEL beam in Fig. 3A is due to the absence of matter (cavity), we cut the LiF sample layer by layer near the expected cavity to a depth of a few tens of micrometers from the surface (see Methods for details)” According to this sentence,**

the LSM measurements are just verified up to a few tens on micrometers, but in Fig 4 D the maximum measured depth is more than 700 μm . Which one is correct? Additionally, the final depth of the drilling hole is not verified. The authors also should add some SEM micrographs at positions several 100 μm deep in the material. Those micrographs might be added to the supplementary material.

Reply: Thank you, corrected. We would like to note here that the sample has been milled up to the depth of only a few 10s of microns to verify the formation of voids. Milling samples more than to 1-mm depth would take weeks or months of using focused ion beam is practically impossible. However, the confocal microscope images presented in Fig.3 demonstrate the dark spot in the beam axis which is experimental evidence of the void formation up to the depth of more than 1,100 μm .

Thus, we combine two methods: (1) FIB and (2) Confocal microscope - to confirm the presence of a long cavity inside the crystal. FIB allows you to examine samples to a depth of several tens of micrometers, while a confocal microscope can look at the entire depth of the sample. However, each of these methods has its drawbacks: FIB has a limited depth of observation, and a confocal microscope, although it can see blackness, cannot always determine whether it is a void. Nevertheless, thanks to the combination of the two methods, we can be sure that the voids really exist and reach a depth of more than 1000 micrometers, which is confirmed by our measurements and complex of simulations.

4. If the final depth of the holes is more than 1 mm, shouldn't they are visible in an optical microscope in transmission through the side?

Reply: It is difficult (one might even say it is impossible for our experimental conditions). The LiF sample has transverse dimensions of the order of cm and a cavity diameter of 1 μm . To see the cavity in an optical microscope, the sample must be cut around the cavity in the form of a rod with a diameter \sim mm (which is practically impossible without destroying the cavity by cracking the sample). However, due to the fact that the LiF crystal is a fluorescent medium, we have a distribution of color centers (F_2 , F_3^+) within the entire sample (essentially a 3D distribution of the X-ray beam). Thus, monitoring the internal structure of the cavity with a non-destructive method — using a confocal laser scanning microscope by recording the intensity of a fluorescent signal within the beam image.

Multi-method simulations:

5. The authors should also discuss the absorption process as due to the excitation of the material a decrease in the optical penetration depth might be expected, as for usual ultrafast laser radiation in the VIS range.

Reply: There is a fundamental difference between laser radiation in the visible range and hard X-rays. Visible radiation has a frequency lower than the plasma frequency in the metal. In the case of dielectrics - the material is transparent, there is a phenomenon of optical breakdown. The dielectric begins to absorb visible radiation after breakdown. Whereas in X-rays the situation is fundamentally different - the material absorbs immediately = regardless of intensity = no optical breakdown.

The laser intensity of $\sim 3 \cdot 10^{18}$ W/cm² is well below the intensity required to generate nonlinear absorption effects in LiF. Before the numerical modelling of the experiment, which (simulations) took years to complete, estimates were made. These are estimates for the physical model you are writing about.

Using the provided in manuscript the maximal absorbed energy density of 895 kJ/cm³ (the highest one in our experiment) it is easy to estimate the number of 9 keV photons absorbed in a unit volume. This concentration of absorbed photons is about 0.62 per a cubic nanometer, which is about 100 time smaller than the concentration of Fluorine atoms in LiF crystal at normal conditions (61 atoms/nm³).

F (Fluorine) with rather deep K-shell (900 eV) dominates in absorption. One 9 keV photon is absorbed at one F-atom. This means that the maximal probability that the photon will encounter an already ionized fluorine atom is about 1%, which can be neglected.

We conclude that (1) the consideration of hollow ions, (2) absorption saturation, and (3) the increase of the d_{att} length due to this saturation all need not be considered under our conditions. The approximation with linear absorption and tabulated value of d_{att} (Henke Tables) is completely adequate to our formulation of the problem.

According to the above estimates, fluxes with intensities greater than $I_{thrNonLin} \sim 10^{20}$ W/cm² are required for the attenuation length of d_{att} to increase by creating a high order ~ 1 concentration of hollow ions. It should be noted that the rate of Auger processes is high. The hole closure on the K-shell due to Auger processes occurs in femtosecond times. While our duration is 20 fs. Therefore, the estimate of $I_{thrNonLin}$ is the lower limit of the required intensities. In fact, the required intensity is even higher.

More details in the Multi-method simulations section have been added. In particular:

“...Phase transition into a plasma state happens within a femtosecond at the very earlier stage of the pulse. The 20-fs pulse intensity is at the level of 10^{18} W/cm², which is far below the criterion the nonlinear saturable absorption of 9-keV photons to emerge in solids (as example it was demonstrated in works (*PRL 127, 163903 (2021)*; *Nature 5080 (2014)*) for Fe). Due to long attenuation length the maximal absorbed energy density near the sample surface is $\xi_2 = 895$ kJ/cm³ for such laser pulse, which corresponds to the concentration of absorbed photons 0.62 nm⁻³ is about 100 times smaller than the concentration of F atoms in LiF crystal (61 atoms/nm³). For this reason, we consider linear absorption process in our experimental conditions....”.

6. **The authors present three different simulation methods to explain different processes in the formation of the final hole. The authors should discuss the limits of each simulation method to clarify why each method can explain only the selected use case. From my point of view, MD represents the most powerful method, and as a simulation of the complete hole is not a limitation, so why not only MD simulations are used?**

Reply: Since even in large-scale MD setup the sample size is limited by a micrometer, our MD simulations are intended to obtain qualitative description of basic processes leading to formation of cavity in depth of LiF crystal heated to peak temperatures and pressures much lower than studied above with the usage of SPH method. More details in the simulation section have been added. In particular:

- “The aim of the numerical simulation is to relate the energy release of the XFEL beam to the features formed (crater, plug, cylindrical cavity radius, channel floor). Since the formed structure has a complex morphology and a high aspect ratio (see Fig. 3A - Zone 1-3), a quantitative description with a single model and one computational code cannot describe the whole physics of the formation of such an elongated structure. Taking to account our simulation results obtained for various energy densities deposited at the different depths, the LiF sample was divided into the following (see Fig. 3A): (zone 1) where the crater and the substance plug are produced by superposition of the cylindrical rarefaction wave with the unloading waves propagated from the free surface (3D SPH, 2D HD); (zone 2) includes the

most part of long cavity produced by the cylindrical rarefaction wave (2D SPH, MD); (zone 3) where energy deposition is near a threshold of cavity formation. Here the cavity is produced by cavitation in the stretched melt and the cavity floor is formed by crystallization of the melt (MD).

We should underline that the phase transition into a plasma state happens within a femtosecond at the very earlier stage of the pulse. The 20-fs pulse intensity is at the level of 10^{18} W/cm², which is far below the criterion the nonlinear saturable absorption of 9-keV photons to emerge in solids (as example it was demonstrated in works^{52,53} for Fe). For this reason, we assume a linear absorption process for our experimental conditions. The formation of the cavity does not depend on a detailed description of the almost isochoric formation of the plasma in a very short time (~ fs), but only on the profile of the input energy. The stage of formation of the hot zone with plasma (i.e. the processes of ionization, deceleration of fast electrons and radiative transfer) are not important for the description of the formation of the cavity, since the radius of the hot zone is known from the experiment and we use the density of deposited energy (not incident) to determine the initial pressure and temperature from the equation of state for modeling the motion of matter leading to the formation of the cavity. In the following simulations, we used the distribution of the value of ξ inside the LiF crystal, as shown in Fig. 3C.”

- MD simulation sections: ...“Since even in large-scale MD setup the sample size is limited by a micrometer, our MD simulations are intended to obtain qualitative description of basic processes leading to formation of cavity in depth of LiF crystal heated to peak temperatures and pressures much lower than studied above with the usage of SPH method. At such low energy deposition the rarefaction wave may not be high enough to produce a cavity during wave action.”

Discussion:

7. **Most paragraphs do not discuss the results but provide instead an outlook for further investigation or a comparison with other researchers. Therefore, the section should revised to represent a discussion of the results.**

Reply: We decided to discuss the results obtained immediately after providing the experimental/modelling data and to compare them with the data from the literature. We believe that this makes it easier for the reader to understand the material. We consider the "Discussion" section as a conclusion where we describe the prospects for using the obtained data. Nevertheless, we have added more details to the "Discussion" section and expanded the discussion of the results in the "Results" section.

Reviewer #3 (Remarks to the Author):

NCOMMS-24-27119 “Formation of high-aspect-ratio nanocavity in LiF crystal using a femtosecond of x-ray FEL pulse” by Makarov et al.

Makarov et coworkers propose a method of volumetric ablation and nanostructuring of materials using ultrashort (20 fs) hard x-ray pulses at 9 keV provided by the FLASH source in Hamburg. The focused x-ray pulse can generate high aspect ratio nanostructures in the volume of a dielectric LiF sample. A range of theoretical continuum and atomistic models are used to simulate the formation of the cavity and the surrounding pressure effects. Claim is made that the

technique can be used to nanostructure bulk metallic materials where no alternatives for volume ablation exists. The modification process based mainly from dimensional analysis and electron microscopy, and supported by theoretical models is associated with shock induced amorphization and cavitation.

Altogether this is a very solid scientific works, where the interaction of short x-ray pulses for nanoprocessing is becoming highly relevant. Demonstrating their importance for a range of structures in materials is appealing and interesting. The discussion is pertinent and insightful, with sometimes a dispersion in the analysis coming from the different models that obscures a clear conclusion on the nanocavitation process and the surrounding amorphization.

Reply: Thank you for encouraging comments!

Nevertheless, some of the claims leave the impression of being rather speculative or at least far-fetched. I will indicate below some

- 1. The manuscript is scarce in details concerning matter excitation or ionization. It is expectable that saturation of x-ray absorption in the FEL micro-spot occurs, given the high dose. No further details are given about the ionization mechanisms, collisional processes, inner-shell decay or scattering, and how these affect the energy deposition profile. No information is given on how fast energy couples to the material at the electronic and molecular level, and how fast phase transitions may occur. I feel that such a discussion may be important to state the level of energy density as the main ansatz for the simulation.**

Reply: Dear reviewer, the authors thank you for this comment. Using the provided in manuscript the maximal absorbed energy density of 895 kJ/cm^3 (the highest one in our experiment) it is easy to estimate the number of 9 keV photons absorbed in a unit volume. This concentration of absorbed photons is about 0.62 per a cubic nanometer, which is about 100 time smaller than the concentration of Fluorine atoms in LiF crystal at normal conditions (61 atoms/nm^3).

F (Fluorine) with rather deep K-shell (900 eV) dominates in absorption. One 9 keV photon is absorbed at one F-atom. This means that the maximal probability that the photon will encounter an already ionized fluorine atom is about 1%, which can be neglected.

We conclude that (1) the consideration of hollow ions, (2) absorption saturation, and (3) the increase of the d_{att} length due to this saturation all need not be considered under our conditions. The approximation with linear absorption and tabulated value of d_{att} (Henke Tables) is completely adequate to our formulation of the problem.

According to the above estimates, fluxes with intensities greater than $I_{\text{thrNonLin}} \sim 10^{20} \text{ W/cm}^2$ are required for the attenuation length of d_{att} to increase by creating a high order ~ 1 concentration of hollow ions. It should be noted that the rate of Auger processes is high. The hole closure on the K-shell due to Auger processes occurs in femtosecond times. While our duration is 20 fs. Therefore, the estimate of $I_{\text{thrNonLin}}$ is the lower limit of the required intensities. In fact, the required intensity is even higher.

More details in the Multi-method simulations section have been added. In particular:

“...Phase transition into a plasma state happens within a femtosecond at the very earlier stage of the pulse. The 20-fs pulse intensity is at the level of 10^{18} W/cm^2 , which is far below the criterion the nonlinear saturable absorption of 9-keV photons to emerge in solids (as example it was demonstrated in works (*PRL 127, 163903 (2021)*; *Nature 5080 (2014)*) for Fe). Due to long attenuation length the maximal absorbed energy density near the

sample surface is $\xi_2 = 895 \text{ kJ/cm}^3$ for such laser pulse, which corresponds to the concentration of absorbed photons 0.62 nm^{-3} is about 100 times smaller than the concentration of F atoms in LiF crystal (61 atoms/nm^3). For this reason, we consider linear absorption process in our experimental conditions....”.

2. The attenuation depth (at 1/e) of 475 μm and the existence of a cavity at more than 1000 μm in depth may suggest that at surface the energy density is about one order of magnitude above the cavitation threshold. Below the cavity end one notices (Fig 3) modification that is not pressure related but merely a phase transformation. Is the possible saturation expected to give a different energy density profile with less peak energy density and larger widths as reported here based solely on geometrical consideration? A study of the threshold of the cavity process with different energies down to the threshold (and not in a single pulse) could illuminate the energy evaluation (according to the estimation a cavity should be formed for input energies in the 1 μJ range, four times above the modification threshold given in Ref. 25). I note also the remarkable stability of the cavity along propagation (1000 μm) despite the variation of the energy density for more than an order of magnitude (Fig. 3A).

Reply: Dear Ref, Thank you for your subtle comments on the text. Let's answer step by step.

2.1. The attenuation depth (at 1/e) of 475 μm and the existence of a cavity at more than 1000 μm in depth may suggest that at surface the energy density is about one order of magnitude above the cavitation threshold.

Reply: That's right, the energy density at the surface of 895 kJ/cm^3 is much greater than the energy density of about 10 kJ/cm^3 at the depth at which the cylindrical cavity ends.

There are three fundamentally different zones along the beam axis. The shortest first zone is located near the sample surface. Its length is of the order of a dozen beam diameter. The longest second zone with length of the order of units of absorption lengths d_{att} . And the transitional zone, where the empty cavity ends. Its length is on the order of a fraction of the absorption length d_{att} .

It should be emphasized that the cavitation phenomenon takes place deep in the thickness of the target, i.e. at the end of the cavity in the third zone. There is no cavitation in the first two zones. In the second zone, the cavity formation is caused by radial rejection of the substance (the concept of zones will be further specified below).

In this case, the substance is pushed radially in the direction away from the beam axis. And crystallizes after its radial shift. The empty cavity arises mainly due to integral irreversible swelling of the target itself. The displacements of the macro-target boundaries are small because the size and volume ($\sim 1 \text{ cm}^3$, see Fig. 1) of the target are much larger than the cavity volume (of the order of 10^{-9} cm^3). Possible densification due to solid-solid phase transition also maybe takes place.

Let's discuss definition of «**cavitation**»: Classical cavitation is a phenomenon with the formation of bubbles filled by vapor in regions where the pressure falls below the pressure of saturated vapor in equilibrium with liquid at a given temperature.

Under our conditions, cavitation happens via formation of voids (empty bubbles without vapor) in the stretched liquid under negative pressure. And the amplitude of such pressure must exceed the rupture/fragmentation threshold of resistance of the liquid to the tensile stress, for which the evaporation into the void is not required. In this case voids/bubbles are formed randomly in a volume, due to thermo-fluctuations in density of the stretched liquid.

In our case, the cavity in zone two is not formed by such cavitation of bubbles. It is formed in an axisymmetric flow directed radially from the beam axis.

Cavitation, in the spirit of the definitions given to this phenomenon above, takes place in the third zone, where radial flow is relatively slow and the temperature of melting is not so high as in the second zone. It is a slow process governed by cooling rate of melt. It is caused by the formation of a channel with liquid inside the rigid solid matrix around the channel — the solid walls surrounding confined liquid are rigid.

In this case, the unusual cavitation occurs due to an extremely large density jump during melting/crystallization in ionic salts - up to 25% of the volume. Accordingly, during cooling and crystallization of the liquid phase, a sharp volume reduction (shrinkage) occurs and cavitation-type ruptures in the freezing liquid inside quasi fixed (quasi rigid) solid walls are possible, see MD simulations.

About the three-zone classification: There are three qualitatively different zones along the beam axis.

In the upper zone, the material motion is determined by the interaction between two different rarefaction waves -- the first is a hemispherical rarefaction wave propagating from the free surface of sample, the second is a rarefaction wave travelling in radial direction behind the diverging cylindrical shock. The depth of this zone into the target thickness is of the order of a dozen beam diameters at the surface; the diameter is 410 nm. Here the flow depends on time and two spatial coordinates – the 2D flow.

At the top of the second zone, the local absorbed energy per unit length along the axis is approximately the same as in the first zone. The matter is that the length equal to ~10 beam (~5-10 um) diameters is much less than the absorption length of 475 μm. But in the 2nd zone the rarefaction wave coming from the 400 nm diameter spot at the free surface of a target is not significant. It is not significant both in amplitude (three-dimensional attenuation when propagating from the spot to distances greater than the spot diameter) and in time delay relative to the processes initiated by a cylindrical divergent shock wave at a depth greater than a dozen beam diameters.

Indeed, the speed of light is "infinite". In the second zone, the significant changes occur in times for which the rarefaction wave travels distances of the order of the beam diameter. Whereas the wave from the surface must travel many diameters to get here (this is said to explain the lag of the rarefaction wave running from the surface into the volume).

Third zone is associated with lower energy deposition (but above the threshold of cavity formation) in the depth of cavity (near the cavity floor), where the radial rarefaction wave itself is too weak to cause formation of voids in the melt. Please see detailed discussion in Sec. MD simulations of cavity formation.

Taking into account the above, we have updated the text about the division into zones in line 335: «...Taking to account our simulation results obtained for various energy densities deposited at the different depths, the LiF sample was divided into the following (see Fig. 3A): (zone 1) where the crater and the substance plug are produced by superposition of the cylindrical rarefaction wave with the unloading waves propagated from the free surface (3D SPH, 2D HD); (zone 2) includes the most part of long cavity produced by the cylindrical rarefaction wave (2D SPH, MD); (zone 3) where energy deposition is near a threshold of cavity formation. Here the cavity is produced by cavitation in the stretched melt and the cavity floor is formed by crystallization of the melt (MD)...».

2.2. “Below the cavity end one notices (Fig 3) modification that is not pressure related but merely a phase transformation.”

Reply: The density of the absorbed energy decreases toward deeper into the target volume. Rapidly absorbed energy increases the pressure if the rate of growth of absorbed energy 20

fs far exceeds the hydrodynamic time scale d_{beam}/c_s – diameter of a beam divided to speed of sound. Pressure increase generates material motion. On the other hand, an increase in energy causes melting if the energy density is above the melting threshold.

2.3. “Is the possible saturation expected to give a different energy density profile with less peak energy density and larger widths as reported here based solely on geometrical consideration?”

Reply: It is said above that we can neglect the formation of hollow ions with high accuracy. Therefore, we can use the linear absorption/attenuation length $d_{\text{att}} = 475$ μm .

The question remains whether the cascade of secondary electrons will not expand the radius of the energy release region. That is, the radius r_{eff} at which the thermalization of secondary electrons occurs. The path length of electrons decreases rapidly as the kinetic energy of a free electron decreases. Therefore, the most dangerous are electrons with energies on the order of keV.

So, how large is the increase in the radius of the energy deposit r_{eff} compared to the radius r_{beam} of a 9 keV photon beam?

It is very important to emphasize the following.

The excess of radius r_{eff} over radius r_{beam} does not depend on the beam intensity.

In the experiment, the LF is used exactly as a radius detector by the glow of the color centers. Small intensities are used to avoid strong deformations and damage to the detector. This is how the beam radius was determined in our previous articles: *J Synchrotron Radiat.* 2023 Jan 1;30(Pt 1):208–216. doi: 10.1107/S1600577522006245 and *Opt. Express* 31, 26383–26397 (2023).

That is, the beam radius r_{beam} is unknown to us. For the beam radius we take the effective radius r_{eff} . This is the radius that includes the broadening due to the cascade of secondary electrons.

Also, simulation by Monte-Carlo package not included into the paper gives for the radial spreading of the cascade generated by 9 keV photons the values ~ 100 -150 nm.

On the other hand, there is a law of conservation of energy. The broadening of the photon beam energy release radius could affect our results following from the physical model and numerical simulations only if the broadening radius turned out to be larger than the cavity radius. The cavity radius in zone 2 has been measured experimentally and is approximately 350 nm, see Fig. 3.

Supplementary argument. The calculation in the adopted physical approximation agrees with the data on the radial size of the cavity and its extent in depth.

2.4. A study of the threshold of the cavity process with different energies down to the threshold (and not in a single pulse) could illuminate the energy evaluation (according to the estimation a cavity should be formed for input energies in the 1 μJ range, four times above the modification threshold given in Ref. 25).

Reply: Ref. 25 is *Opt. Express* 31, 26383–26397 (2023). This work focuses on much smaller intensities ~ 1 -10 kJ/cm^3 than those ~ 1000 kJ/cm^3 analyzed in this paper.

Dear reviewer, the article *Opt. Express* 31, 26383 (2023) is devoted to the study namely of threshold effects. But these are effects on the free surface of our target. On the surface, as mentioned above in our reply to 2.1, the interaction of two different rarefaction waves determines the picture. One of them comes from the free surface, the other is radial from the beam axis.

The paper *Opt. Express* 31, 26383 (2023) can only be applied to a qualitative assessment of the situation in zone three of the long channel at huge energy densities of $\sim 10^3$ kJ/cm^3 at the entrance to the single crystal. These energy densities are 100-1000 times above the threshold considered in paper *Opt. Express* 31, 26383 (2023).

To understand the physical situation in zone three, an original MD simulation was performed. It turned out that in order of magnitude the thresholds at the surface and in the depth are comparable. This is physically clear. But the picture in zone three, i.e. in the end zone of the hollow cylinder turned out to be complex and physically meaningful.

Namely:

- 1) A large jump of volume reduction in ionic salts (LiF refers to them) during solidification.
- 2) Solidification of a liquid in a channel with solid low-deformable walls.

The last part of comment 2

2.5 I note also the remarkable stability of the cavity along propagation (1000um) despite the variation of the energy density for more than an order of magnitude (Fig. 3A).

Reply: It is impossible to bend the channel. Unabsorbed photons propagate along the normal to the surface, and the channel diameter changes slightly along its length (on the scale of the Fig.3B it is not so noticeable). The channel diameter changes along its length due to the angular divergence of the beam. At a length z of 920 μm , the diameter doubles - see formula (3) in Methods ($\lambda = 0.138 \text{ nm}$; $M^2=3$, $r_0 = 205 \text{ nm}$):

$$r_{beam}^2(z) = r_0^2 \left[1 + \left(\frac{z}{z_R} \right)^2 \right] = r_0^2 \left[1 + \left(\frac{\ln(2) z \lambda M^2}{2\pi r_0^2} \right)^2 \right]$$

The length of doubling the diameter is:

$$z^2 = 2\text{Pi}(205 \text{ nm})^2 / (\text{Log}[2] \lambda M^2) = 920 \text{ um.}$$

3. **A discussion based on the energy density required to kick-start a process is given at page 9. The main argument is dimensional, saying that melting or cracking occur in regions where estimated local energy density based on pulse profile is too small so the process is not driven by local energy but by shock pressure coming from the central area, triggered by fast isochoric heating (the argument is that the melting area has a radius of 0.6 um with respect to the 0.3um at 1/e2 expected from the x-ray pulse, low energy case). A full proof of such discussion would require the measured metrology of the X-ray pulse at the measured plane as scattering and absorption saturation may locally modify the profile with respect to Fig. 2C. Secondly, it is important to highlight the process dynamics; for example, a relaxation of a lower density amorphous phase upon cooling may create (due to the density mismatch) type of stress components that will generate spallation or cracking of the adjacent areas. This will provide a hint to the state of the material when it suffers the modification, as its evolution may depend on it. Third, thermal transport may be active as long as the sample is hot and heat may propagate in the surroundings. More arguments may be needed to support the statement in addition to the crude geometrical evaluation.**

Reply: Dear reviewer, this is the second part of your comment #2. Thank you for these opinions.

Two aspects are addressed in this commentary. (1) The melting radius and the cracking radius are discussed. (2) The intensity profile of the photon beam in the plane perpendicular to the beam axis.

It is also said that it is necessary to describe the dynamics of the process which is initiated by the absorption of the beam energy.

Response.

We consider very high intensities with energy $\sim 1000 \text{ kJ/cm}^3$, which are more than two decimal orders of magnitude above the modification, cracking and melting thresholds. Under these conditions, the radiatively initiated flow goes through several different phases/stages in time as the shock expands and its amplitude weakens down to the melting/cracking stages. We are now talking about zones one and two along the axis of the cylinder. The main achievement of the paper is the proof of the formation of a long empty cylinder - cavity.

In zone 2, such an empty\hollow cylinder is formed before the considered flow weakens to the cracking stage.

Additional subsidiary achievements are:

(a) the reasoning about melting and cracking in zone 1.

(b) on the cruciform character of the damage outside the hole of the hollow cylinder.

Circles (a) and (b) are external to the cavity hole. Observation (b) is an interesting observation. It turns out to be typical for a single crystal with a packing of cubic character. Here is a reference to the paper [*Laser Pulse Induced Dislocation Structure in Ionic Crystals, From the book Volume 66, Number 2 August 16, A.V. Gobbunov , E. M. Nadgornyi and S. N. Valkovskii, <https://doi.org/10.1515/9783112501269-006>*], see Fig below:

We are sure that the reasoning about the phenomena of melting and cracking due to the action of a divergent shock wave weakening as it propagates is correct. Indeed, the shock wave in cylindrical geometry attenuates in a power-law manner as a function of radius and time.

Detailed studies of the photon beam intensity wings are devoted to our previous works. They are based on experimental measurements. It is shown that the attenuation of the beam intensity in the plane perpendicular to the beam axis follows Gauss's law. It is a very fast strong attenuation proportional to the exponent: $\exp(-r^2/r_{\text{beam}}^2)$. Of course, this law of attenuation is much faster than the power-law rate of attenuation.

The power-law attenuation of a cylindrical shock has been considered here [*Attenuation and inflection of initially planar shock wave generated by femtosecond laser pulse, Optics & Laser Technology, Volume 152, August 2022, 108100*]:

$$p \propto 1/r_{\text{sw}}^{(3/4)},$$

$$l \propto r_{sw}^{(1/4)},$$

$$l/r_{sw} \rightarrow 1/r_{sw}^{(3/4)}.$$

Here “ p ” is amplitude of a shock, “ r_{sw} ” is instant radial position of a shock front, “ l ” is thickness of a shock compressed layer behind the shock front.

That is, the amplitude of the shock wave is attenuated in a power-law manner with an exponent of 3/4. This attenuation rate is much weaker than the attenuation with radius according to the Gaussian law.

In addition, as it propagates, the compressed shell behind the wave front becomes thinner and thinner.

- 4. The suggestions several times in the text of TPa pressure levels as well as polymorphic transformations are not really supported here, though these appear as key elements of the discussion.**

Reply: TPa pressure level is the Young modulus of diamond, representing the limit of pressure for material transformation and the following studies of their unusual physical and chemical properties with diamond anvil cell. This pressure limit corresponds to the energy density achieved by concentrating 1 μJ of laser energy in 1 μm^3 volume of material which is easily reached by optical and now – x-ray ultrashort laser pulses. We mention this TPa pressure to demonstrate that laser-induced microexplosion leads to new non-equilibrium phase transformation pathways and opens up a range of new material end phases with novel material properties.

- 5. The authors suggest, quite understandably, a strong interest for metals however their use case is a dielectric where optical Bessel Gauss pulses can produce similar high aspect cylindrical geometries for nanocavities that do not require MJ/cm^3 . This puts forward some inconsistencies in the interpretation of nanocavity formation with optical and x-ray pulses, though dimensionally cavities and surrounding amorphous or stressed regions are similar. The dynamic behaviour suggested here seem equally different from the one reported with optical pulses and this issue should be addressed. Phase transitions, mismatch in amorphous and crystalline densities, and heat transport were also suggested to explain modifications observed post mortem, without invoking high levels of mechanical shock.**

Reply: We have recently experimentally demonstrated using Bessel-Gaussian beams that cylindrical 2-D expansion of shock wave is more efficient way to achieve shock wave pressure above 1 TPa and the resulted density is ~ 1.2 time higher than that previously reached with 3-D microexplosion with conventional Gaussian beams (*Opt. Express* 30, 6016 (2022)). This demonstration is directly relevant to the presented here microexplosion experiments with focused XFEL pulses.

- 6. A comprehensive and relevant simulation package is used to analyse the hydrodynamics and MD aspects of the process starting with the evaluated input energy density profile (hence the need to accurately confirm it). Timescales in the range of ns are found (Fig. 4C) while sub-ns times are given in Juodkazis et al. PRL 2006 and several tens of ns are given in (Nguyen et al. Ultrafast Sci. 2024). Such discrepancies may perhaps be explained by the nature of the material and irradiation conditions but I feel correlations should be made as similar processes are discussed. What would be the theoretical energy threshold for cavity formation when all heating, structural transformation, and relaxation processes are accounted**

for and how the phase transformation of the material influences the process? Some elements of the discussion are given at Fig. 6 but the times of several ps shows apparently some inconsistencies with Fig. 3.

The MD simulation brings into the discussion the cooling of the hot melt and its influence on the cavitation, while so far the discussion was dominated by shock launch. Perhaps a conclusive and convergent discussion coming from the different codes used here may put forward the main common elements, with a focus on the nanocavity formation; the key element of interest.

Reply: The reviewer indicated the articles for quartz irradiated with an 800 ns laser. In our experiment we had hard X-rays. The difference from optics we added in the introduction section. In general, difference in the time scales of observed processes (in our article and Juodkazis et al. PRL 2006 and Nguyen et al. *Ultrafast Sci.* 2024) is related to the size of the irradiated spot. Hydrodynamic evolution (characteristic) time is determined by hot spot size (heating area) divided the speed velocity c_s . The speed of sound does not depend much on solid materials, so the main spread in time scales is determined by the size of the hot spot (for large spots, it is necessary to consider large hydrodynamic times, as for the case of optics in the articles mentioned).

Also, in the article by *Nguyen et al Ultrafast Sci. 2024* the energy input was small, the rarefaction wave was weak to create a cavity. However, heating led to a phase transition to the amorphous quartz with HIGHER density than normal one. Thus, negative pressure arose, and many nanobubbles are independently formed, but a unique long cavity is not formed in opposite to our case. The proposed mechanism of nanopore formation in article is not realized in our conditions (since amorphous LiF most likely does not exist as a stable phase). However, the nanovoids can be also formed in LiF below the cavity floor. We have added a new statement in our manuscript (see MD simulation Section): “It is of interest to observe in Supplemental Video 2 that below the cavity floor the fast homogeneous crystallization results in formation of nanovoids between solid grains by reason of much lower density of molten LiF compared to solid LiF (by 25%).”

To make clear on different mechanisms of cavity formation (depending on deposited energy density) we have added in the revised manuscript the following classification: “Taking to account our simulation results obtained for various energy densities deposited at the different depths, the LiF sample was divided into the following (see Fig. 3A): (zone 1) where the crater and the substance plug are produced by superposition of the cylindrical rarefaction wave with the unloading waves propagated from the free surface (3D SPH, 2D HD); (zone 2) includes the most part of long cavity produced by the cylindrical rarefaction wave (2D SPH, MD); (zone 3) where energy deposition is near a threshold of cavity formation. Here the cavity is produced by cavitation in the stretched melt and the cavity floor is formed by crystallization of the melt (MD).”

7. **To the stress distribution, there may be several elements leading to such a distribution. As a general note, radial transport will create perpendicular stress components and similar polarization patterns.**

Reply: Such specific cross-shaped stress distribution is not formed in our SPH hydrocode simulations if material is considered as an ideal elastic-plastic matter. The application of a damage model (with stress relaxation produced by a damage parameter) for material represented by SPH particles with lattice packing (rather than homogeneous liquid-like packing) is essential to reproduce the cross-shaped pattern. Thus, the symmetrical radial flow of material itself cannot create such pattern. Also see the revised manuscript with a new statement on 425 line: “It was confirmed by a test run with a homogeneous liquid-like

packing of SPH particles, which provides an angle-independent damage and pressure distributions.”

The LiF crystal as a lattice with angle-dependent strength responds to shock compression by an angle-dependent damage distribution, which results in the corresponding stress relaxation. Our atomistic MD simulation demonstrates such response.

Perhaps the radial transport of heat was meant. But such transport may create only angle-independent stress field if the initial temperature distribution was axial-symmetric. We think that this is the case since the spatial profile of X-ray pulse was axial-symmetric with high precision.

In conclusion, this is a very interesting and original report. It reads nicely even though sometimes the discussion gains in complexity when surface and volume processes are discussed and mixed.

Nevertheless, I feel that at this point, despite the high level of scientific discussion, the comprehensive analyses and theoretical simulations, and the originality of the irradiation conditions, the manuscript does not illuminate new physical aspects related to the interaction of short energetic pulses and matter in the cavitation range. Cavities and nanobubbles were reported in dielectric materials in several occasions and their conditions analysed, with perhaps conclusions that may also be relevant to this work and also alternative formation scenarios. The cylindrical relaxation geometry was found optimal for generating high aspect ratio nanochannels up to 1:10000. What appears notable here is the high energy density values, that may potentially create specific relaxation conditions for the X-ray pulse. Their validation is therefore critical to the scenarios proposed here and to the conclusive section of the manuscript. The claimed extreme conditions need therefore further experimental insight to be confirmed (with arguments in addition to the dimensional analysis using a pulse profile propagation in “vacuum”), as similar post-mortem damage and cavity appearances can be achieved optically at lower energies (energy density at least one order of magnitude lower), with similar sections for cavities and surrounding amorphous regions. The authors should demonstrate why secondary processes such scattering and absorption saturation as well as fast diffusion cannot increase the interaction section. The similarities and differences with respect to the cavitation induced by optical (Bessel) beams either in terms of energy density or process dynamics should be outlined. At the same time elements of beam metrology in interaction at high doses should be given in support of the dimensional analysis. This will help to clearly indicate the shock scenario as most probable, opposite to a transformation related to local energy density and not driven by propagating mechanical waves. References to extreme polymorphic transformations should be demonstrated otherwise they appear speculative in the context.

Therefore, despite the interest of the report, I cannot give a positive recommendation for publication in Nature Communications. However, I may reconsider if direct and convincing experimental proof of high shock levels is given together with the associated polymorphic transformations, transient or permanent, including the validation of the energy density levels.

Reply: Thank you for your high appreciation of our work and detailed review. This work presents the first, to our knowledge, study of mm-long nanochannel formation by a single high brilliance X-ray free electron laser pulse. Intense X-ray pulses generated by free-electron-lasers offer new avenues for formation of nanochannels for nanofluidic systems in any solid material. We tried to give convincing answers to all questions, as well as expand on the theses in the text of the article. Thanks to your comments, we have significantly revised the manuscript. In particular, the discussion section put forward an idea about polymorphs as a direction for future research and experiments. Since we did not study this issue in detail in our experiment, we removed the text from the introduction. However, we

have shown that shock wave with TPa-level pressure is formed in this experimental conditions that leads to «radial extrusion» of matter. Such pressure can lead to the formation of void surrounded by a compressed shell with novel metastable polymorphic structures. The major difficulty in studying such polymorphs produced around a tiny optical focal spot is the very low, of the order of femtograms, quantity of material, which makes it extremely difficult to detect and characterise their unusual structural and electronic properties. Here, we demonstrate that the volumes of shockwave affected material can be increased by >1000 times by hard X-ray pulses from free-electron lasers.

Miscellaneous.

The addition of drawings in Fig;2 makes it difficult to evaluate what the authors correlate with melting. The SEM appearance is not by itself the strongest argument for melting.

Reply: The SEM measurements shown in Figure 2 (at two observation angles – 45 and 90 degrees) demonstrate us clear signs of melting in the purple area: (1) a smooth inner surface of the circle area (Fig.2a,d) (the surface was levelled by capillary tension), which also led to influxes of matter (see e.g. Fig. 2d – molten substance); a cylindrically symmetrical region within the purple fill, Fig.2b,e (in the case of spallation and destruction, broken chips are always observed, as can be seen outside the purple region). It is also consistent with simulation data.

We would like to thank the referees for the work they performed and their impression of the manuscript.

To make proofreading easier, we have highlighted in **red** all differences with the previous versions. The following document contains our answers (written in **blue**) to the referee's comments (written in **black**).

On behalf of all co-authors,

S. Makarov

REVIEWER COMMENTS

“I note at the same time a technologically-g geared introduction which seems far-fetched for XFEL as similar performances can be reached easier with optical beams.”

Response:

The introduction specifically highlights the limitations of optical beams in nanochannel formation. The main limitation is the possibility of creating such structures only in optically transparent materials. An additional complication is introduced by the nonlinear light-matter interaction with a strong dependence of energy deposition on fluctuating local carrier density. This aspect limits the possibility of controlled creation of nanochannels. The fundamental novelty of our study lies in the first-ever demonstration of millimetre-long nanochannel formation with an aspect ratio exceeding 1,000, using a single X-ray free-electron laser pulse with photon energy in the 10 keV range, which absorption is linear and well-controlled in contrast to optical photons. Importantly, virtually all materials are transparent at these photon energy levels, enabling the formation of smooth nanochannel even in materials that are opaque to optical wavelengths—this represents a fundamentally new opportunity in the field.

“Moreover, resemblances of X ray damage with optical damage on surface (exfoliation present on ionic crystals in fs laser ablation experiments) and in the bulk (cavities are expected for tight-focused excitation in cylindrical and spherical geometries) raises questions about the expected extreme behaviors. Are extraordinary processes to be expected at 0.3-0.9MJ/cm³ levels given that sub-micron cavities can be obtained with a tenth of deposited energy?”

Response:

We agree that the shallow sub-micron craters with an aspect ratio up to ~10 could be obtained at the surface with much smaller deposited energy density due to the ejection of the material close to the sample surface. However, the formation of long nanochannels with much higher aspect ratio does not involve the removal of material away from the channel due to the very fast quenching of material in ultrashort laser-matter interaction in the bulk of the material. Instead, the extraordinary processes observed at deposited energy densities of 0.3–0.9 MJ/cm³ result from a fast transition into a hot highly pressurized state of matter. This state is characterized by solid-state density and ionized plasma temperatures of several tens eV, commonly referred to as warm dense matter (WDM).

The resulting radially expanding shock wave pressure exceeds the material's Young's modulus and leads to the formation of a void in the material with compressed nanochannel walls – see the first description of these extraordinary processes in Refs.[32,33] in the

manuscript. Moreover, it was experimentally demonstrated (Appl.Phys.Lett. 110, 161907) that the high-aspect ratio nanochannel formation is due to the shock wave compression of the material, which is fundamentally different to the shallow sub-micron cavities formed at the tenth of deposited energy due to formation of an ablated shallow crater on the surface. In those experiments the array of nanochannels formed in WDM conditions using an optical Bessel beam resulted in the reduced volume of the sample by ~22-29%, while the sample mass was reduced only by 0.3-0.8%. Thus, our results provide compelling evidence that WDM conditions can be achieved and that high aspect ratio nanochannels can be formed using hard ultrashort XFEL pulses.

“Without clear indications (for example, the absence of experimental confirmation of sub-nanosecond shock and cavity dynamics), the physics here becomes less singular. Theories and models about materials under constraints were discussed in various occasions. The demonstration I suggested is the structural proof of matter following extreme evolution with a potential dynamic view of the process. Despite the compelling comments and numerical methods, such a proof was not given and statements remain partly speculative. The novelty remains utilization of XFEL source (notwithstanding the proof-of-concept material also transparent at optical wavelengths). I have the feeling that strength lies more in the perspectives that were proposed. I therefore maintain the initial recommendation.”

Response:

We fully acknowledge that experimental confirmation of shock wave propagation and cavity formation dynamics would provide direct validation of the physical processes involved in nanochannel formation. In response to this, we are actively developing such experiments using a pump-probe technique, where both pump and probe pulses are in the X-ray spectral domain. These experiments are at the forefront of current technological capabilities, requiring ultra-precise alignment, high temporal resolution, and significantly enhanced X-ray detection sensitivity. Despite these challenges, ongoing advancements in XFEL capabilities – including higher photon energy and increased pulse intensity, make experimental studies of nanochannel formation dynamics increasingly feasible in the very near future.

At this stage, the simulation presented in our manuscript remains the only viable approach to uncovering these dynamics. The choice of a transparent material in our experiments is specifically justified by the ability to visualize the nanochannel beyond its initial few tens of microns using a confocal optical microscope – currently the only method available for assessing the full length of the nanochannel.

Following your recommendations, we have carefully revised our manuscript to remove any potentially speculative statements. All modifications are highlighted in the manuscript file.

Dear Reviewer,

Thank you very much for your constructive comments and for your kind words describing the discussion in our manuscript as “very good and particularly rich.” Below we provide our responses to the specific points you raised.

Q1: The main question, in my opinion, is whether the manuscript demonstrates physical transformations not achievable by other means or illuminates unique mechanisms of transformation. These points remain unanswered to me.

Reply: The key experimental result of our work is the first demonstration of the delivery and localization of X-ray laser energy down to sub-micron dimensions over a millimetre-long focal region within a solid material, achieving an energy density of up to 0.9 MJ/cm³. This is accomplished through an unprecedented concentration of X-ray energy over a long focusing distance.

The resulting energy density significantly exceeds the Young’s modulus of the material and leads to the formation of high-aspect-ratio hollow nanochannels, a transformation that is both experimentally characterised and theoretically substantiated in the manuscript.

To our knowledge, this is the first observation of nanochannel formation in the bulk of a solid material induced by a single ultrashort X-ray pulse. The distinctiveness, reproducibility and localisation of this transformation clearly demonstrate a novel physical mechanism of nanochannel formation uniquely accessible through X-ray free-electron laser (XFEL) irradiation.

These findings open a new path for high-aspect-ratio nanochannel formation in a wide range of solids, including inaccessible to optical lasers. This capability creates new opportunities for exploring transport phenomena, nanoscale fluid dynamics, and fundamental physics and chemistry principles that govern ultrafast nonlinear flow of liquid in nanochannels.

Q.2: Choice of material: If this is a key point, why not demonstrating the technique on materials that have no other alternatives?

Reply: We respectfully disagree with the Reviewer’s interpretation of the manuscript’s key point. Our principal contribution is not the specific choice of material but the experimental demonstration and theoretical validation of the ability to concentrate energy from a single XFEL pulse to levels approaching ~MJ/cm³ within a solid, – much above the yield strength of any material. We should also underline that the demonstrating the universality of the channel formation method in all materials using an XFEL, as you requests, cannot be achieved in a single beam time experimental slot. Having conducted experiments and extensive simulations for LiF crystal, we can assert that this method works in this material. Furthermore, based on other experimental works and an understanding of the basic physical mechanisms common to all materials, we suppose that the method is universal for dielectrics, semiconductors and metals.

We believe that the article contains significant results that encourage the continuation of this kind of research by other laboratories. That’s how science works. A step forward is being taken by one group. This spark initiates the global scientific community to take the next steps.

Q3. Why not demonstrate the technique on materials that have no other alternatives?

Reply: We addressed this point in our prior response (Round 2) ^{*}, but we elaborate further here.

We selected LiF as the demonstration material primarily because it is optically transparent and permits a straightforward correlation between nanochannel length and the 9-keV X-ray linear absorption depth. This greatly simplifies optical diagnostics (e.g., Fig. 3) and supports accurate modeling and interpretation of the results.

While LiF is the focus of this initial study, our aim is to establish a clear and well-understood system for establishing and validating the proposed mechanism. This lays the groundwork for future work exploring a broader range of materials, particularly optically opaque and technologically relevant solids. As example, it was already demonstrated in work [52 from the main article] that a similar structure in the form of an extended cavity with a diameter of 4 μm (aspect ratio 1:10) was produced in the Si semiconductor after irradiation with focused XFEL pulse ($d_{FWHM} = 1 \mu\text{m}$, $\tau = 20 \text{ fs}$) with a photon energy of 10 keV and an absorbed energy density $\xi = 430 \text{ kJ cm}^{-3}$ per pulse. We should point out that the described mechanisms apply to any type of material where a high energy density region is rapidly created. In this region, distinctions between materials are erased, and a very high-pressure plasma forms. The surrounding cold material then moves like a fluid under this pressure. This also indicates the universality of the hollow channel formation mechanism along almost its entire length, with the exception of the channel's very end (zone 3). At this point, the material's strength and thermophysical properties might lead to differences in how various materials behave. Therefore, we believe that similar hollow structures will be observed when irradiating metals with XFELs, for which experimental results are currently lacking.

Q.4: Mechanism and the discussion of extreme mechanics involved in the void formation. Alternative scenarios (e.g., Yao et al.) exist for nanostructuring without shock compression.

On the mechanism and discussion of extreme mechanics: ... Experimental evidence is required to support the shock-based mechanism.

Reply:

We respectfully disagree with the interpretation that the formation of high-aspect-ratio nanovoids can generally proceed without shock compression. In our scenario, the formation of the hollow nanochannel is a direct result of the following sequence of events:

1. The localized and absorbed X-ray energy density ($\sim \text{MJ/cm}^3$) is several times higher than the Young's modulus of LiF ($\sim 65 \text{ GPa}$).
2. This energy input ionizes the material in the focal volume, forming a confined plasma.
3. The plasma rapidly expands, launching a strong shock wave that compresses the surrounding solid material.
4. The rarefaction wave, which follows the shock front (as required by mass conservation), creates the void, resulting in the observed hollow channel.

To move atoms away from their equilibrium positions inside a solid and form a void, the energy density must exceed the mechanical strength of the material, which is characterized by the Young's modulus. This is consistent with our calculations and modeling.

^{*} Our reply from Round 2 discussion: *“The choice of a transparent material in our experiments is specifically justified by the ability to visualize the nanochannel beyond its initial few tens of microns using a confocal optical microscope – currently the only method available for assessing the full length of the nanochannel.”*

In the study by Z. Yao et al. with optical Bessel beam indicated by the Reviewer (<https://doi.org/10.1364/OE.26.021960>) the estimated energy density in PMMA is ~ 13 GPa[†], which is significantly higher than the bulk modulus of PMMA of ~ 2 - 3 GPa, and its yield strength 0.05 - 0.1 GPa (see [<https://doi.org/10.1016/j.rinp.2016.05.004>], [[https://www.google.com/url?sa=t&source=web&rct=j&opi=89978449&url=https://www.sciencedirect.com/topics/agricultural-and-biological-sciences/polymethylmethacrylate%23:~:text=3DPolymethylmethacrylate%2520\(PMMA\)%2520is%2520a%2520strong,and%252050%2520MPa%2520%255B44%255D.&ved=2ahUK EwiUjLmQw7SOAxWgHRAIHq7BJkQzsoNegQIJhAa&usg=AOvVaw0BFxnd8eFUNltenQY-DyIOJ](https://www.google.com/url?sa=t&source=web&rct=j&opi=89978449&url=https://www.sciencedirect.com/topics/agricultural-and-biological-sciences/polymethylmethacrylate%23:~:text=3DPolymethylmethacrylate%2520(PMMA)%2520is%2520a%2520strong,and%252050%2520MPa%2520%255B44%255D.&ved=2ahUK EwiUjLmQw7SOAxWgHRAIHq7BJkQzsoNegQIJhAa&usg=AOvVaw0BFxnd8eFUNltenQY-DyIOJ)]). Therefore, even in that case, the formation of voids involves a shock-driven mechanism.

Our theoretical model and numerical simulations confirm the central role of shock wave dynamics in the formation of high-aspect-ratio voids observed in our experiments. We are confident that for the extreme energy densities generated by XFEL pulses, shock wave dynamics appear to be the dominant process.

Conclusion:

This article is the fruit of the work of many co-authors and many laboratories over a long period of time. The experimental results were completed by 2021. It took three years to build a physical model and numerical simulation. Finally, our efforts to publish the article have been ongoing for more than a year (since April 2024).

We believe that our responses address the Reviewer's concerns and clarify both the novelty and significance of our results. The manuscript presents the first demonstration of bulk nanochannel formation via localized X-ray absorption at ultrahigh energy densities, supported by experimental evidence and physical modeling. These findings establish ultrafast X-ray irradiation as a fundamentally new tool for nanostructuring.

We respectfully ask the Reviewer to reconsider their evaluation and recommend the manuscript for publication in *Nature Communications* without further delay.

Kind regards,

Sergey Makarov (on behalf of the authors)

[†] In the paper by Z. Yao et al, the laser pulse energy at the input to the optical system is $E = 60 \mu\text{J}$. By considering very conservatively the efficiency of SLM and losses in the optical system forming the Bessel beam as ~ 20 - 50% , the average length of the channel formed as $L = 500 \mu\text{m}$ and the radius of the channel as $r = 1 \mu\text{m}$, the energy density in the plasma column is $P = 0.35 \times E / (L \times \pi r^2) = 0.35 \times 6 \times 10^5 \text{J} / (500 \mu\text{m} \times \pi \times (1 \mu\text{m})^2) \approx 1.3 \times 10^4 \text{J}/\text{cm}^3 = 13 \text{GPa}$.